# UNIHM: UNIFIED DEXTEROUS HAND MANIPULATION WITH VISION LANGUAGE MODEL

**Zhenhao Zhang**[1,2*], **Jiaxin Liu**[1*], **Ye Shi**[1,2†], **Jingya Wang**[1,2†]

`{zhangzhh2024,liujx2024,shiye,wangjingya}@shanghaitech.edu.cn`

[1] ShanghaiTech University [2] InstAdapt

## ABSTRACT

Planning physically feasible dexterous hand manipulation is a central challenge in robotic manipulation and Embodied AI. Prior work typically relies on object-centric cues or precise hand-object interaction sequences, foregoing the rich, compositional guidance of open-vocabulary instruction. We introduce UniHM, the first framework for unified dexterous hand manipulation guided by free-form language commands. We propose a Unified Hand-Dexterous Tokenizer that maps heterogeneous dexterous-hand morphologies into a single shared codebook, improving cross-dexterous hand generalization and scalability to new morphologies. Our vision language action model is trained solely on human-object interaction data, eliminating the need for massive real-world teleoperation datasets, and demonstrates strong generalizability in producing human-like manipulation sequences from open-ended language instructions. To ensure physical realism, we introduce a physics-guided dynamic refinement module that performs segment-wise joint optimization under generative and temporal priors, yielding smooth and physically feasible manipulation sequences. Across multiple datasets and real-world evaluations, UniHM attains state-of-the-art results on both seen and unseen objects and trajectories, demonstrating strong generalization and high physical feasibility. Our project page at https://unihm.github.io/.

## 1 INTRODUCTION

Dexterous hand manipulation involves perceiving, grasping, and reconfiguring objects in complex environments. Generating and understanding diverse, long-horizon, and physically feasible dexterous-hand manipulation sequences are critical for advancing robotic capabilities in humanoid-centric applications. Such human-like interactions are ubiquitous in the real world and underpin fine-grained, complex tasks for embodied agents.

Real-world manipulation tasks require sequential, contact-rich control that couples semantic intent (*what to do*) with precise geometry and physics (*how to do it*). Conventional methods (Wang et al., 2022; Xu et al., 2023; Li et al., 2025) either take object-centric inputs and optimize static grasp pose or transfer human-object interaction (HOI) video to fixed sequences. **Lacking open-vocabulary instruction**, these pipelines cannot guide diverse and complex dexterous hand manipulation.

Recent vision-language approaches have begun to guide static grasping and manipulation by mapping free-form language to grasp representations. SemGrasp (Li et al., 2024b) uses a language-aligned discretization of the grasp space and fine-tunes a multimodal LLM so that text, object cues, and discrete grasp tokens lie in a shared semantic space, yielding language-conditioned static human grasp poses. AffordDexGrasp (Wei et al., 2025a) targets open-set generalization by predicting affordances from language and then generating dexterous grasps, with the primary output still being static pose rather than multi-step interaction. Despite advances in semantic controllability and open-vocabulary coverage, most language-guided approaches focus on generating **static grasp poses**, ignore temporal structure, and therefore fail to produce smooth and rich manipulation sequences.

Building on these observations, we propose **UniHM**, a unified framework for generating dexterous hand-manipulation sequences for both seen and unseen objects under open-vocabulary language in-

---

*Equal contribution;    †The corresponding author.

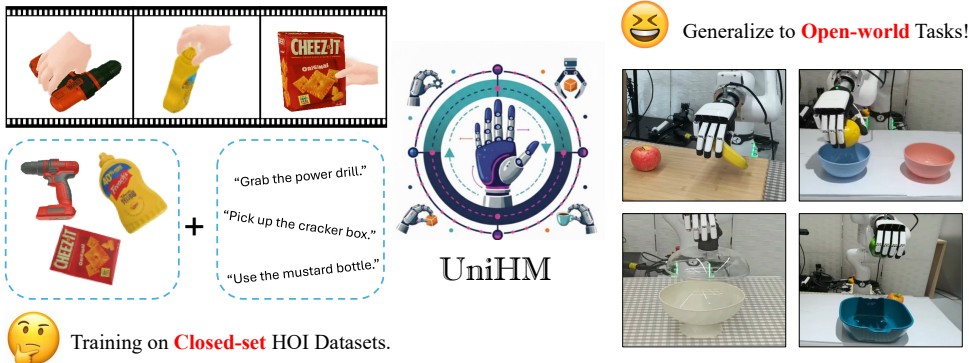

Figure 1: **Overview.** We introduce UniHM, the first unified hand-manipulation framework conditioned on free-form language. UniHM is trained solely on closed-set HOI datasets to follow target trajectories and execute physically feasible interactions, and generalizes to open-world tasks in real-world interactions.

structions. Our approach couples a vision-language model with a Unified Hand-Dexterous Tokenizer and physics-guided dynamic refinement, which together enable two key capabilities: **1) Dynamic language-guided manipulation.** Prior methods typically either generate only static grasp poses or lack open-vocabulary language understanding; in contrast, we progressively train both the trajectory planner and the DexHand VLM, allowing the system to perform dynamic dexterous-hand manipulation along arbitrary planned trajectories with substantially improved temporal consistency and scalability. **2) Learning from video.** We leverage diverse, scene-rich human video data that contains abundant information about both environments and interactions(Zeng et al., 2025), enabling the model to synthesize human-like grasping poses while achieving strong generalization across a wide range of manipulation scenarios. Extensive real-world cross-embodiment experiments further demonstrate the effectiveness and generalization ability of this training paradigm. By integrating these components, **UniHM** achieves state-of-the-art performance and exhibits strong generalization across hand morphologies, object categories, task horizons, and linguistic complexity.

Our contributions summaries are as follows:

- **Unified Dexterous Hand Manipulation.** We introduce **UniHM**, the first unified, language-conditioned framework for dynamic dexterous hand manipulation beyond static grasps directly from images and open-vocabulary instructions.

- **Morphology-Agnostic Codebook.** We introduce a unified VQ token codebook with cross-hand consistency that maps heterogeneous hand kinematics into one discrete action lattice and decodes tokens into hand-specific joint trajectories, which enables direct token reuse and transfer across robotic and anthropomorphic hands.

- **Physics-Guided Dynamic Trajectory Optimization.** We employ a tailored energy-based refinement that fuses a generative prior shaping feasible pose manifolds, a temporal prior enforcing smooth velocity–acceleration profiles and time-aware consistency, and contact-aware dynamic trajectory optimization to optimize the generation result.

- **Generalization without Teleoperation.** Our framework eliminates the dependency on expensive teleoperation data by learning dexterous manipulation skills from human videos. This paradigm achieves robust generalization to unseen scenes and instructions, significantly lowering the barrier to developing dexterous manipulation systems.

## 2 RELATED WORK

### 2.1 DEXTEROUS GRASP GENERATION.

Research on dexterous grasping largely follows two lines. **Language-free** pipelines advance dexterous grasping without text by exploiting geometry, simulation, or video demonstrations. UniDexGrasp (Xu et al., 2023) learns from point clouds using a two-stage scheme that first proposes diverse grasp poses and then executes a goal-conditioned policy for stable lift, thereby generalizing across many categories. DexGraspNet (Wang et al., 2022) synthesizes and validates at scale via differentiable force-closure and collision checks to supervise grasp policies. DexMV (Qin et al., 2022) builds an imitation pipeline that converts human videos into robot demonstrations through pose estimation, retargeting, and demonstration translation for dexterous manipulation. **Language-guided** dexterous grasping aligns natural language with grasp spaces using discretization or affordance cues; for example, SemGrasp (Li et al., 2024b) learns a language-aligned discrete representation and uses a VLM to produce semantically consistent static dexterous grasp poses. DexGYS (Wei et al., 2024) introduces a language-guided dataset and model that maps commands to dexterous grasp poses. AffordDexGrasp (Wei et al., 2025a) leverages a generalizable, instructive affordance representation to guide open-set dexterous grasp generation. While these methods improve semantic controllability and open-vocabulary coverage, they remain pose-centric and typically generate **static poses** rather than sequence-level hand manipulation.

### 2.2 VISION LANGUAGE MODEL FOR MANIPULATION

Generative vision-language model approaches increasingly cast manipulation as sequence prediction conditioned on vision and language. MotionGPT (Zhu et al., 2025) treats human motion as a language via VQ tokenization and unified text-motion modeling for sequence generation. HOIGPT (Huang et al., 2025) extends token-based generation to long 3D hand-object interaction, learning a bidirectional mapping between text and HOI sequences. OWG (Tziafas & Kasaei, 2024) composes VLM-guided referring segmentation, grounded grasp planning, and contact-aware ranking for zero-shot grasping rather than full sequence synthesis. ReKep (Huang et al., 2024) converts language/vision into relational keypoint constraints and solves actions via hierarchical optimization instead of an autoregressive policy. DexGrasp Anything (Zhong et al., 2025b) complements VLM pipelines with a physics-aware diffusion generator and the largest dexterous-grasp dataset to date, improving feasibility through explicit physical constraints. Multi-GraspLLM (Li et al., 2024a) aligns point-cloud and text features to generate language-guided grasp poses across multiple robotic hands within a single framework. Taken together, these generative models advance instruction following but predominantly target **Digital Hand, low-DoF grippers, or static grasp poses**; explicit sequential dexterous-hand manipulation with feasibility remains limited, motivating our hand-agnostic tokenizer and physics-guided refinement.

## 3 METHOD

We propose **UniHM**, a unified framework for dexterous hand manipulation. UniHM first annotates large-scale hand-object interaction data, then employs a vision language model coupled with a Unified Hand-Dexterous Tokenizer to synthesize manipulation sequences, and finally applies physics-based optimization to ensure physical feasibility.

### 3.1 AUTO DATA ANNOTATION

**Open-vocabulary Language Annotation.** We annotated dexterous hand manipulation sequences with GPT-4o (Hurst et al., 2024). For each sequence, we provide keyframes as visual context, specifically the first and last frames and the three frames preceding first contact, and the model returns five distinct open-vocabulary natural language instructions.

**Dexterous Hand Retargeting.** For HOI sequences to dexterous hand manipulation sequences transfer, we first apply Dex-Retargeting (Qin et al., 2023) to map MANO poses onto five dexterous robot hands (Shadow hand, Allegro hand, SVH hand, Leap hand and Panda hand). We then perform

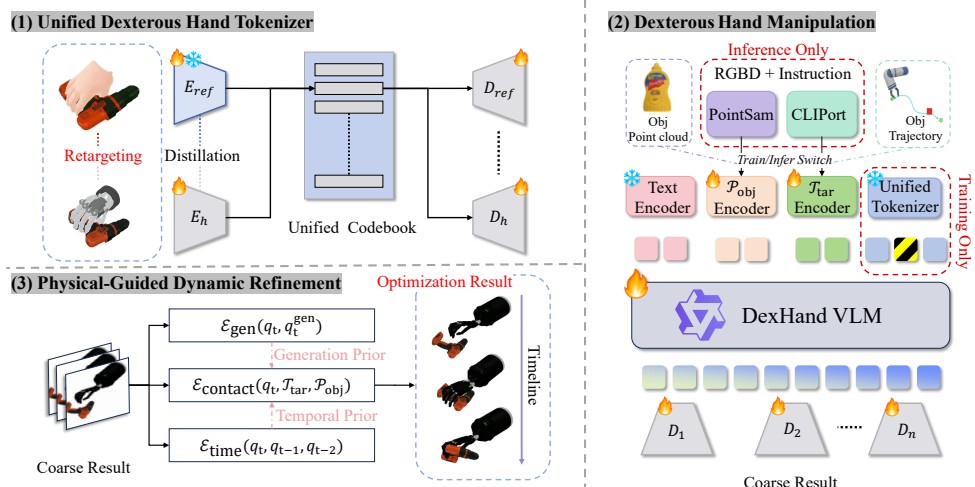

Figure 2: **Pipeline.** UniHM converts open-vocabulary instructions and RGB-D inputs into executable dexterous-hand trajectories via three stages: (1) morphology-agnostic motion tokenization; (2) language-guided generation that fuses text, perception, and token history to produce manipulation token sequences; and (3) physics-aware decoding with smoothness/contact priors for feasible, stable execution.

energy-based physical optimization to enforce feasibility, yielding physically consistent dexterous hand manipulation sequences.

## 3.2 UNIFIED HAND-DEXTEROUS TOKENIZER

**Morphology-Agnostic Codebook.** We discretize dexterous hand poses and short motion segments across heterogeneous hand morphologies with a shared VQ-VAE (Van Den Oord et al., 2017) codebook. Let $E_h$ and $D_h$ denote the encoder and decoder for hand type $h$ (e.g., MANO or a specific robot hand), and let the global codebook be $\mathcal{Z} = \{\mathbf{e}_k\}_{k=1}^K$ with $\mathbf{e}_k \in \mathbb{R}^{d_z}$. Given a sequence chunk $\mathbf{x}^{(h)}$, we define the encoder output $\mathbf{z}_e^{(h)} = E_h(\mathbf{x}^{(h)})$ and the vector-quantization operator $Q(\cdot)$ that returns an index $c \in \{1, \ldots, K\}$:

$$c = Q\left(E_h\left(\mathbf{x}^{(h)}\right)\right) = \arg\min_{k \in [K]} \left\|\mathbf{z}_e^{(h)} - \mathbf{e}_k\right\|_2^2, \tag{1}$$

and the quantized representation is $\mathbf{z}_q^{(h)} = \mathbf{e}_c$. The reconstructed sequence chunk $\hat{\mathbf{x}}^{(h)}$ is then

$$\hat{\mathbf{x}}^{(h)} = D_h\left(\mathbf{z}_q^{(h)}\right) = D_h\left(\mathbf{e}_c\right). \tag{2}$$

So that every encoder $E_h$ maps to the same discrete index space and every decoder $D_h$ realizes the token within its own morphology.

**Scalable Cross-Morphology Training.** To align representations across diverse hand types, we train our model in a staged, scalable manner. We first establish a reference encoder-decoder pair ($E_{\text{ref}}$, $D_{\text{ref}}$) for a specific hand morphology, along with the shared codebook $\mathcal{Z}$.

When integrating a new hand morphology, instead of direct non-differentiable token alignment, we first align the new encoder's latent space with the reference encoder via knowledge distillation. This bypasses the gradient discontinuity of the quantization step. The distillation objective is:

$$\mathcal{L}_{\text{distill}} = \|E_{\text{new}}\left(\mathbf{x}_{\text{new}}\right) - E_{\text{ref}}\left(\mathbf{x}_{\text{ref}}\right)\|_2^2, \tag{3}$$

where $E_{\text{new}}$ is the new encoder and $E_{\text{ref}}$ is the pre-trained reference encoder. $\mathbf{x}_{\text{new}}$ and $\mathbf{x}_{\text{ref}}$ are corresponding hand sequences for the new and reference hands, respectively, obtained through retargeting.

After the new encoder is aligned, it is integrated into the full VQ-VAE training pipeline. The new encoder and its corresponding decoder are then fine-tuned with a combined objective that includes reconstruction and codebook-related terms. For a new hand type $h$, the training objectives are:

$$\mathcal{L}_{\text{rec}} = \left\| \mathbf{x}^{(h)} - D_h\left(\mathbf{z}_q^{(h)}\right) \right\|_2^2 \tag{4}$$

$$\mathcal{L}_{\text{vq}} = \left\| \text{sg}\left[\mathbf{z}_e^{(h)}\right] - \mathbf{z}_q^{(h)} \right\|_2^2 + \beta \left\| \mathbf{z}_e^{(h)} - \text{sg}\left[\mathbf{z}_q^{(h)}\right] \right\|_2^2, \tag{5}$$

where $\text{sg}[\cdot]$ denotes the stop-gradient operator and $\beta > 0$ is a commitment weight. The total training objective for a new morphology is a combination of these terms, as well as the distillation loss for encoder alignment.

**Unified Hand Pose Translation.** Because the encoders are aligned to a shared latent space and the decoders are mapped to the unified codebook, translating a pose from hand $i$ to hand $j$ is straightforward. We simply encode the source pose and decode using the target hand's decoder:

$$\hat{\mathbf{x}}^{(j)} = D_j\left(\mathbf{e}_{Q\left(E_i\left(\mathbf{x}^{(i)}\right)\right)}\right), \tag{6}$$

where $\mathbf{x}^{(i)}$ is the source hand pose and $\hat{\mathbf{x}}^{(j)}$ is the retargeted pose for the target hand.

## 3.3 Dexterous-Hand Manipulation with VLM

**Network structure of VLM.** Given the scarcity of HOI and dynamic dexterous-hand data and the high cost of collecting it, training large multimodal language models (e.g., 7B or 13B parameters) is data-inefficient and yields limited performance in this regime(Zeng et al., 2026; Wang et al., 2026; He et al., 2025; Zhong et al., 2025a). We therefore adopt **Qwen3-0.6B** as the base model, whose scale enables stable convergence on HOI and dexterous-hand datasets. To compensate for the lack of dynamic manipulation data while retaining strong visual grounding, We adopt a decoupled architecture that separates scene perception from HOI sequence generation. A CLIPort-style vision module consumes RGB-D images and language to infer target trajectories, and an MLP-based trajectory encoder serializes these targets for the VLM. The main VLM focuses on instruction-conditioned token generation with a progressive masking curriculum, and the unified tokenizer maps tokens to hand-specific poses across different robotic hands. At inference time, only the CLIPort perception head is adapted to distribution shifts, which improves data efficiency and robustness while keeping the HOI generator unchanged.

**Unified Dexterous Manipulation.** We first use a CLIPort model to take an RGB-D image and its corresponding instruction (e.g., "grasp the bottle into the box") as input. The model then decodes a target execution trajectory $\mathcal{T}_{\text{tar}}$. This process can be formally described as

$$\mathcal{T}_{\text{tar}} = \text{CLIPort}(I_{rgb-d}, T_{instruction}). \tag{7}$$

The target trajectory is denoted by $\mathcal{T}_{\text{tar}} = \{\mathcal{T}_{\text{tar}}^1, \ldots, \mathcal{T}_{\text{tar}}^K\}, \quad \mathcal{T}_{\text{tar}}^i \in \text{SE}(3)$, where $K$ is the length of the total sequence.

We leverage RGB-D data to reconstruct the scene point cloud and then use Point-SAM to segment the object point cloud $\mathcal{P}_{\text{obj}}$ with the corresponding semantics.

$$\mathcal{P}_{\text{obj}} = \text{PointSAM}(I_{rgb-d}, T_{instruction}). \tag{8}$$

$P_{\text{obj}}$ denotes the point cloud of the target object. Specifically, $P_{\text{obj}} \in \mathbb{R}^{3 \times l}$ is a set of $l$ 3D coordinates $(x, y, z)$, where $l$ is the number of points in the point cloud.

Next, the initial hand pose is encoded by $E_j(\cdot)$ and concatenated with $\mathcal{T}_{\text{tar}}$, $\mathcal{P}_{\text{obj}}$, and text token $\mathcal{T}$. This combined input is then fed into a Vision Language Model (VLM) to obtain a code sequence in the VQ-VAE latent space. Finally, we use the decoder of the target hand, $D_h(\cdot)$, to decode the corresponding q-positions:

$$\hat{Q}_{pos} = D_h(\text{VLM}(E_j(Qpos_0), \mathcal{T}_{\text{tar}}, \mathcal{P}_{\text{obj}}, \mathcal{T})). \tag{9}$$

**Training Stage.** When fine-tuning the main model, we input the real object sequence trajectory as a target trajectory into the large model. To enable the model to understand the spatial continuity of

hand poses, we initially input the ground truth hand poses and the VQ-VAE as an encoder into the model. As training progresses, we use masking to randomly occlude a portion of the hand poses, replacing the masked poses with a unified, learnable token, until all hand poses are masked (Liu et al., 2019; Oquab et al., 2023).

The masking process can be expressed as:

$$\mathcal{Q}_{\text{masked}} = M \odot E(\mathcal{Q}) + (1 - M) \odot \mathcal{T}_{\text{mask}}, \tag{10}$$

where $\mathcal{Q}$ is the ground truth hand pose sequence, $E$ is the VQ-VAE encoder, $M$ is a binary mask, and $\mathcal{T}_{\text{mask}}$ is the learnable token.

**Inference Stage.** Our training and inference pipelines differ by design. During training, we condition the model on ground-truth target trajectories and object point clouds, enabling it to reliably generate physically feasible dexterous-hand manipulation sequences that follow the specified targets. At inference, a separate CLIPort module estimates these quantities from RGB-D observations, decoupling spatial perception from hand–object interaction. This modularization allows the main model to focus solely on HOI sequence generation. A practical advantage is robustness to environmental shift: when the scene distribution changes, we fine-tune only CLIPort, which is smaller and less data-hungry, rather than the entire model.

### 3.4 Physical-guided Dynamic Refinement

To bridge the gap between the generated grasping trajectory $\mathcal{Q}_{\text{gen}}$ and the object point cloud $\mathcal{P}_{\text{obj}}$ under the target pose trajectory $\mathcal{T}_{\text{tar}}(t)$, we formulate a posterior optimization. We solve a spatio-temporally regularized, frame-by-frame Gauss–Newton problem that enforces physical plausibility while preserving the semantic intent of the generated actions. We proceed frame-by-frame: at time $t$, we optimize $q_t$ while treating $q_{t-1}^{\text{opt}}$ and $q_{t-2}^{\text{opt}}$ as fixed from previous steps.

**Contact Energy.** Let $s_i(q_t)$ be the $i$-th fingertip position from forward kinematics in the world frame. We transform it into the object frame using $\mathcal{T}_{\text{tar}}(t)^{-1}$ and query the nearest neighbor $(\mathbf{p}_i, \mathbf{n}_i)$ on $\mathcal{P}_{\text{obj}}$, where $\mathbf{n}_i$ is the local surface normal. We define a signed point-to-plane distance (Grant et al., 2012):

$$d_i(q_t) = \mathbf{n}_i^{\text{T}} \left( \mathcal{T}_{\text{tar}}(t)^{-1} s_i(q_t) - \mathbf{p}_i \right), \tag{11}$$

and an asymmetric, smooth penalty that is continuous and slope-matched at $d = 0$, which is essential during the optimization:

$$f(d) = \begin{cases} \frac{\alpha}{2} d^2, & d \geq 0 \text{ (outside)} \\ \frac{\alpha}{k^2} \left( e^{-kd} + kd - 1 \right), & d < 0 \text{ (inside)} \end{cases}, \tag{12}$$

where $\alpha > 0$ and $k > 0$ are scale parameters. Stacking per-fingertip residuals yields:

$$r_{\text{contact},i}(q_t) = \sqrt{2 \lambda_c f(d_i(q_t))}, \quad \mathcal{E}_{\text{contact}}(q_t) = \frac{1}{2} \left\| r_{\text{contact}}(q_t) \right\|_2^2 = \lambda_c \sum_{i \in \text{fingers}} f(d_i(q_t)), \tag{13}$$

with $\lambda_c > 0$ controlling the contact strength.

**Generative HOI Prior.** To preserve the intent of the generative model, we penalize deviations from the generated configuration $q_t^{\text{gen}}$:

$$\mathcal{E}_{\text{gen}}(q_t, q_t^{\text{gen}}) = \frac{1}{2} \left( q_t - q_t^{\text{gen}} \right)^{\text{T}} \mathbf{W}_{\text{gen}} \left( q_t - q_t^{\text{gen}} \right), \tag{14}$$

where $\mathbf{W}_{\text{gen}} \succ \mathbf{0}$ is a symmetric positive-definite weighting matrix. We use the weighted norm shorthand $\|x\|_{\mathbf{W}}^2 \triangleq x^{\text{T}} \mathbf{W} x$ below.

**Temporal Prior.** We regularize first- and second-order temporal differences to ensure smooth, coherent motion:

$$\mathcal{E}_{\text{time}}(q_t, q_{t-1}^{\text{opt}}, q_{t-2}^{\text{opt}}) = \frac{1}{2} \left\| q_t - q_{t-1}^{\text{opt}} \right\|_{\mathbf{W}_{\text{vel}}}^2 + \frac{1}{2} \left\| (q_t - q_{t-1}^{\text{opt}}) - (q_{t-1}^{\text{opt}} - q_{t-2}^{\text{opt}}) \right\|_{\mathbf{W}_{\text{acc}}}^2, \tag{15}$$

where $\mathbf{W}_{\text{vel}} \succ \mathbf{0}$ and $\mathbf{W}_{\text{acc}} \succ \mathbf{0}$ (often $\mathbf{W}_{\text{vel}} = \lambda_{\text{vel}} I$, $\mathbf{W}_{\text{acc}} = \lambda_{\text{acc}} I$). For $t < 2$, we use zero-velocity/acceleration or the generated states as boundary priors, meaning that $q_{-2}^{\text{opt}} = q_{-1}^{\text{opt}} = q_0^{\text{gen}}$.

Table 1: Main Result on DexYCB. The arrow pointing to the right means closer to the GT.

| | Method | MPJPE↓ | FOL↓ | FPL↓ | FID↓ | Diversity → |
|---|---|---|---|---|---|---|
| | GT | - | - | - | - | 125.53 |
| **Seen** | TM2T(Guo et al., 2022) | $85.33^{\pm3.41}$ | $36.57^{\pm1.46}$ | $24.10^{\pm0.96}$ | $54.83^{\pm2.19}$ | $37.12^{\pm1.48}$ |
| | MDM(Tevet et al., 2023) | $88.06^{\pm3.52}$ | $33.40^{\pm1.34}$ | $23.06^{\pm0.92}$ | $52.33^{\pm2.09}$ | $33.95^{\pm1.36}$ |
| | FlowMDM(Barquero et al., 2024) | $82.75^{\pm3.31}$ | $31.24^{\pm1.25}$ | $21.55^{\pm0.86}$ | $48.05^{\pm1.92}$ | $61.25^{\pm2.45}$ |
| | MotionGPT3(Zhu et al., 2025) | $74.80^{\pm2.99}$ | $28.76^{\pm1.15}$ | $19.32^{\pm0.77}$ | $43.35^{\pm1.73}$ | $\mathbf{72.51}^{\pm2.90}$ |
| | **Ours** | $\mathbf{61.40}^{\pm1.93}$ | $\mathbf{23.14}^{\pm0.65}$ | $\mathbf{12.15}^{\pm0.24}$ | $\mathbf{31.24}^{\pm1.02}$ | $39.62^{\pm0.66}$ |
| **Unseen** | TM2T(Guo et al., 2022) | $94.22^{\pm3.77}$ | $37.25^{\pm1.49}$ | $27.03^{\pm1.08}$ | $55.94^{\pm2.24}$ | $31.25^{\pm1.25}$ |
| | MDM(Tevet et al., 2023) | $93.05^{\pm3.72}$ | $39.04^{\pm1.56}$ | $25.89^{\pm1.04}$ | $55.13^{\pm2.21}$ | $29.0^{\pm1.16}$ |
| | FlowMDM(Barquero et al., 2024) | $86.13^{\pm3.45}$ | $32.67^{\pm1.31}$ | $24.09^{\pm0.96}$ | $51.33^{\pm2.05}$ | $58.21^{\pm2.33}$ |
| | MotionGPT3(Zhu et al., 2025) | $77.93^{\pm3.12}$ | $30.55^{\pm1.22}$ | $21.48^{\pm0.86}$ | $46.14^{\pm1.85}$ | $\mathbf{75.84}^{\pm3.03}$ |
| | **Ours** | $\mathbf{63.56}^{\pm2.08}$ | $\mathbf{27.29}^{\pm0.43}$ | $\mathbf{13.06}^{\pm0.43}$ | $\mathbf{41.03}^{\pm1.65}$ | $42.70^{\pm1.19}$ |

The per-frame objective is the sum of all terms:

$$\mathcal{E}_t(q_t, q_t^{\text{gen}}, q_{t-1}^{\text{opt}}, q_{t-2}^{\text{opt}}) = \mathcal{E}_{\text{contact}}(q_t) + \mathcal{E}_{\text{gen}}(q_t, q_t^{\text{gen}}) + \mathcal{E}_{\text{time}}(q_t, q_{t-1}^{\text{opt}}, q_{t-2}^{\text{opt}}), \quad (16)$$

and the sequence energy is $\mathcal{E}_{\text{total}}(\mathcal{Q}) = \sum_{t=0}^{T} \mathcal{E}_t(q_t, q_t^{\text{gen}}, q_{t-1}^{\text{opt}}, q_{t-2}^{\text{opt}})$.

We linearize $r_{\text{contact}}(q_t)$ and apply Gauss–Newton with Levenberg–Marquardt damping $\lambda \geq 0$ (Yu & Wilamowski, 2018). Let $J_t = \partial r_{\text{contact}}(q_t)/\partial q_t$. The normal equations for the update $\Delta q_t$ are:

$$\left(J_t^{\text{T}} J_t + \mathbf{W}_{\text{gen}} + \mathbf{W}_{\text{vel}} + \mathbf{W}_{\text{acc}} + \lambda I\right)\Delta q_t = -J_t^{\text{T}} r_{\text{contact}}(q_t) - \tilde{\mathbf{W}} \quad (17)$$

$$\tilde{\mathbf{W}} \triangleq \mathbf{W}_{\text{gen}}\left(q_t - q_t^{\text{gen}}\right) + \mathbf{W}_{\text{vel}}\left(q_t - q_{t-1}^{\text{opt}}\right) + \mathbf{W}_{\text{acc}}\left(\left(q_t - q_{t-1}^{\text{opt}}\right) - \left(q_{t-1}^{\text{opt}} - q_{t-2}^{\text{opt}}\right)\right). \quad (18)$$

Here the prior terms enter both as quadratic regularizers on the left and as linear gradients on the right, stabilizing the optimization while respecting the generated intent and temporal smoothness under the object's target pose trajectory $\mathcal{T}_{\text{tar}}(t)$.

## 4 EXPERIMENTS

Our framework comprises three components: (1) we annotate hand-object interaction (HOI) sequences with a vision language model (VLM) and, via retargeting plus physics-based refinement, obtain manipulation sequences for multiple dexterous hands; (2) we train a Unified Hand-Dexterous Tokenizer together with an VLM for text-to-hand manipulation sequence generation; and (3) we apply an energy-based physical refinement and validate feasibility in simulation. Comparative experiments and ablations demonstrate the effectiveness of our approach. All experiments are conducted on NVIDIA A100 GPUs.

### 4.1 DATASET

In our experiments, we evaluate on two of the most widely used datasets. **DexYCB** (Chao et al., 2021) is a multi-view RGB-D dataset of human grasps on YCB objects with precise 3D labels: 582K frames, 1,000 sequences, 10 subjects, 20 objects, and 8 views; benchmarks include 2D/6D object pose, 3D hand pose, and handover. **OakInk** (Yang et al., 2022) A large-scale hand-object interaction repository integrating OakInk-Image (230K multi-view frames from 12 subjects manipulating 100 objects across 32 categories) and OakInk-Shape (3D grasp-pose meshes with affordance labels), plus 50K affordance-aware interactions transferred via Tink. For both **DexYCB** and **OakInk**, we adopt an 80/20 split: 80% for training/validation (seen) and 20% held out as an unseen test set. This protocol enables a rigorous assessment of generalization across seen and unseen objects, trajectories, and interaction patterns for UniHM.

### 4.2 EVALUATION METRIC

To assess the quality, diversity, realism, and physical plausibility of our generated dexterous-hand manipulation sequences, we adopt a multi-pronged evaluation protocol that follows prior work (Wei

Table 2: Main Result on OakInk. The arrow pointing to the right means closer to the GT.

| | Method | MPJPE↓ | FOL↓ | FPL↓ | FID↓ | Diversity → |
|---|---|---|---|---|---|---|
| | GT | - | - | - | - | 147.40 |
| Seen | TM2T(Guo et al., 2022) | $71.08^{\pm 2.84}$ | $91.25^{\pm 3.65}$ | $34.51^{\pm 1.38}$ | $311.90^{\pm 12.48}$ | $277.38^{\pm 11.10}$ |
| | MDM(Tevet et al., 2023) | $67.55^{\pm 2.70}$ | $93.8^{\pm 3.75}$ | $30.06^{\pm 1.20}$ | $285.22^{\pm 11.41}$ | $275.42^{\pm 11.02}$ |
| | FlowMDM(Barquero et al., 2024) | $60.74^{\pm 2.43}$ | $85.43^{\pm 3.42}$ | $26.47^{\pm 1.06}$ | $249.08^{\pm 9.96}$ | $189.54^{\pm 7.58}$ |
| | MotionGPT3(Zhu et al., 2025) | $56.29^{\pm 2.25}$ | $79.24^{\pm 3.17}$ | $23.98^{\pm 0.96}$ | $221.10^{\pm 8.84}$ | $247.10^{\pm 9.88}$ |
| | **Ours** | $\mathbf{52.73}^{\pm 2.08}$ | $\mathbf{72.32}^{\pm 0.55}$ | $\mathbf{19.86}^{\pm 1.38}$ | $\mathbf{204.91}^{\pm 7.64}$ | $\mathbf{165.47}^{\pm 6.30}$ |
| Unseen | TM2T(Guo et al., 2022) | $75.34^{\pm 3.01}$ | $125.33^{\pm 5.01}$ | $45.51^{\pm 1.82}$ | $337.08^{\pm 13.48}$ | $362.08^{\pm 14.48}$ |
| | MDM(Tevet et al., 2023) | $72.90^{\pm 2.92}$ | $112.94^{\pm 4.52}$ | $42.93^{\pm 1.72}$ | $325.58^{\pm 13.02}$ | $354.93^{\pm 14.20}$ |
| | FlowMDM(Barquero et al., 2024) | $65.39^{\pm 2.62}$ | $101.25^{\pm 4.05}$ | $36.14^{\pm 1.45}$ | $298.04^{\pm 11.92}$ | $224.67^{\pm 8.99}$ |
| | MotionGPT3(Zhu et al., 2025) | $61.95^{\pm 2.48}$ | $93.65^{\pm 3.75}$ | $28.25^{\pm 1.13}$ | $272.69^{\pm 10.91}$ | $316.58^{\pm 12.66}$ |
| | **Ours** | $\mathbf{58.62}^{\pm 2.35}$ | $\mathbf{83.27}^{\pm 1.17}$ | $\mathbf{22.87}^{\pm 0.52}$ | $\mathbf{253.41}^{\pm 13.05}$ | $\mathbf{153.28}^{\pm 9.48}$ |

Table 3: Real-World Experiments

| Split | Method | Success Rate | | | |
|---|---|---|---|---|---|
| | | Grab | Pick&Place | Pull&Push | Open&Close |
| Seen | MDM+Dex-Retargeting | 20% | 10% | 0% | 5% |
| | MotionGPT3+Dex-Retargeting | 30% | 15% | 25% | 25% |
| | **Ours** | **65%** | **50%** | **60%** | **55%** |
| Unseen | MDM+Dex-Retargeting | 5% | 0% | 0% | 5% |
| | MotionGPT3+Dex-Retargeting | 45% | 25% | 15% | 20% |
| | **Ours** | **60%** | **35%** | **55%** | **45%** |

et al., 2025a; Zhang et al., 2025; Deng et al., 2025; Ji et al., 2025; Liu et al., 2025; Wang et al., 2025b; Jian et al., 2025; Wei et al., 2025b) and combines quantitative and qualitative measures across five metrics.

- **Physically Feasible.** Mean Per-Joint Position Error (**MPJPE**) for hand joints, Final Position Location Error (**FPL**) and Final Orientation Location Error (**FOL**) for dexterous hand placement and orientation. **Success Rate** means the Real-world grasping success rate.

- **Generation Realism.** Fréchet Inception Distance (**FID**) between real and synthesized. **Diversity,** which measures the variability across different prompts and within outputs from the same prompt

Lower MPJPE, FPL, FOL, and FID indicate higher accuracy and fidelity. Diversity closer to the ground truth indicates a more reasonable generation.

## 4.3 MAIN RESULT

**Comparison with SOTA Methods.** We conduct extensive comparisons against prior state-of-the-art methods(i.e.,TM2T (Guo et al., 2022), MDM (Tevet et al., 2023), FlowMDM (Barquero et al., 2024), MotionGPT3 (Zhu et al., 2025)) on both DexYCB (Chao et al., 2021) and OakInk (Yang et al., 2022). Because prior action-generation baselines lack explicit physical-feasibility guarantees, we post-process their outputs with our physics-guided refinement to ensure a fair comparison. As shown in Table 1 and Table 2, our method consistently outperforms all baselines across both seen and unseen objects.

These results unequivocally affirm our method as cutting-edge, showcasing the robust generalization capability of our VLM framework and physical refinement, thereby substantiating the efficacy of our UniHM synthesis framework in producing sequential hand manipulation sequences for both seen and unseen objects and trajectories guided by open-vocabulary instructions.

**Visualization in Real-world Results.** We conduct real-world evaluations on a dexterous hand across both seen and unseen objects and trajectories. Results show improvements in grasp success rate and grasp quality. As summarized in Table 3, **UniHM** achieves a higher success rate than prior methods, and Fig.D2 presents qualitative visualizations of our real-world grasps.

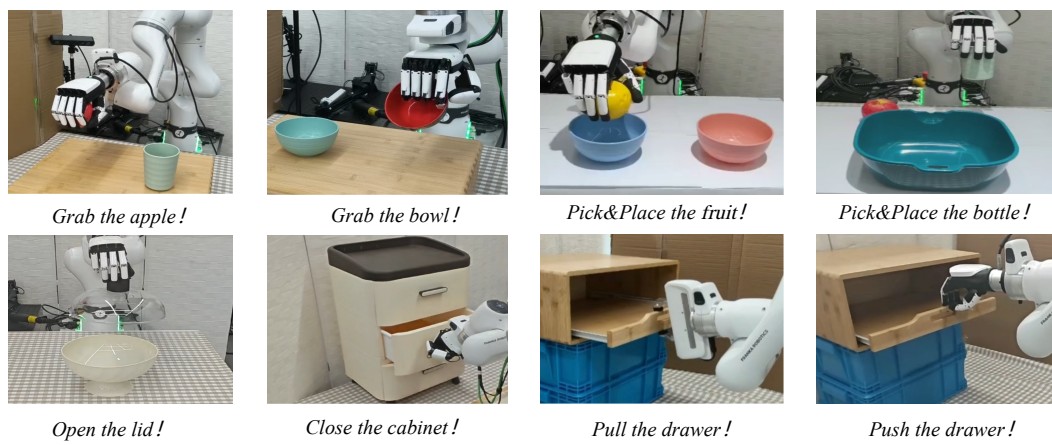

*Grab the apple!*  *Grab the bowl!*  *Pick&Place the fruit!*  *Pick&Place the bottle!*

*Open the lid!*  *Close the cabinet!*  *Pull the drawer!*  *Push the drawer!*

Figure 3: **Real-World Results.** UniHM achieves higher success rates than prior methods on both seen and unseen objects, producing physically consistent and executable real-world manipulations.

Table 4: Ablation Result on DexYCB. The arrow pointing to the right means closer to the GT.

|  | Method | MPJPE↓ | FOL↓ | FPL↓ | FID↓ | Diversity → |
|---|---|---|---|---|---|---|
|  | GT | - | - | - | - | 125.53 |
| **Seen** | w/o Depth Input | $85.47^{\pm3.42}$ | $33.41^{\pm1.34}$ | $20.97^{\pm0.84}$ | $56.36^{\pm2.25}$ | $66.40^{\pm2.66}$ |
|  | w/o Masked Training | $73.41^{\pm2.94}$ | $28.22^{\pm1.13}$ | $14.42^{\pm0.58}$ | $44.87^{\pm1.79}$ | $\mathbf{73.09}^{\pm2.92}$ |
|  | w/o Physical Refinement | $65.78^{\pm2.63}$ | $25.06^{\pm1.00}$ | $15.35^{\pm0.61}$ | $33.57^{\pm1.34}$ | $38.06^{\pm1.52}$ |
|  | **Ours** | $\mathbf{61.40}^{\pm1.93}$ | $\mathbf{23.14}^{\pm0.65}$ | $\mathbf{12.15}^{\pm0.24}$ | $\mathbf{31.24}^{\pm1.02}$ | $39.62^{\pm0.66}$ |
| **Unseen** | w/o Depth Input | $90.12^{\pm3.60}$ | $39.77^{\pm1.59}$ | $21.70^{\pm0.87}$ | $77.38^{\pm3.10}$ | $67.53^{\pm2.70}$ |
|  | w/o Masked Training | $74.63^{\pm2.99}$ | $28.08^{\pm1.12}$ | $17.25^{\pm0.69}$ | $43.09^{\pm1.72}$ | $\mathbf{74.88}^{\pm3.00}$ |
|  | w/o Physical Refinement | $65.39^{\pm2.62}$ | $28.55^{\pm1.14}$ | $16.05^{\pm0.64}$ | $45.06^{\pm1.80}$ | $41.03^{\pm1.64}$ |
|  | **Ours** | $\mathbf{63.56}^{\pm2.08}$ | $\mathbf{27.29}^{\pm0.43}$ | $\mathbf{13.06}^{\pm0.43}$ | $\mathbf{41.03}^{\pm1.65}$ | $42.70^{\pm1.19}$ |

## 4.4 ABLATION STUDY

We conduct controlled ablation studies on **DexYCB** (Chao et al., 2021) to verify the necessity of each module. Each component is essential for **UniHM** to synthesize realistic and physically feasible dexterous hand manipulation sequences; the details can be found in Table 4.

**Masked Training (w/o Masked Training).** We adopt a progressive masking curriculum for language-conditioned sequence generation. Training starts with the teacher forcing the use of both language and ground-truth sequences, then gradually replaces a fraction $p_t$ of ground truth with [MASK]. As $p_t$ increases from 0 to 1, the model relies on language and its autoregressive history; the final stage uses language only, matching inference. This reduces exposure bias and improves sequential stability while retaining strong supervision early on.

**RGB-D or RGB Input (w/o Depth Input).** To better estimate object poses and reconstruct scene point clouds, we adopt RGB-D inputs and infer scene trajectories with a language-conditioned visuomotor module. When RGB-D is replaced by RGB-only, pose estimation and 3D reconstruction degrade substantially.

**Physical Refinement (w/o Physical Refinement).** Physical refinement is a postprocessing step that makes a plausible plan physically valid for dexterous hands. We run a lightweight simulation-based optimization that adjusts poses, contacts, and timing to reduce collisions and slips, enforce joint and torque limits, and improve stability. The objective penalizes penetration, excessive contact forces, and abrupt accelerations while staying close to the original.

## 5 CONCLUSION

We present **UniHM**, a unified framework for synthesizing sequential dexterous hand-manipulation sequences guided by open-vocabulary instructions. UniHM couples a Unified Hand-Dexterous Tokenizer, which utilizes a shared codebook with cross-hand code distillation to form a common discrete action space, with a vision language model for instruction-grounded token generation. It applies a physical-guided dynamic refinement, yielding semantically aligned and physically consistent trajectories across heterogeneous robotic hands. Extensive evaluations of DexYCB and OakInk, along with simulation checks and real-world trials, demonstrate strong generalization to unseen objects and trajectories, and validate the contribution of each component through ablations. While UniHM advances language-conditioned dexterous manipulation beyond pose-centric grasp generation, several challenges remain: reliance on RGB-D perception without tactile or force sensing, simplified energy terms for contact and friction, and limited coverage of bimanual or tool-use scenarios. Future work will incorporate richer contact priors and feedback, scale the unified codebook to more hand morphologies, and close the loop with online adaptation to better handle sequential, contact-rich tasks.

## ACKNOWLEDGEMENT

This work was supported by National Natural Science Foundation of China (62406195, 62303319), ShanghaiTech AI4S Initiative SHTAI4S202404, HPC Platform of ShanghaiTech University, and MoE Key Laboratory of Intelligent Perception and Human-Machine Collaboration (ShanghaiTech University), and Shanghai Engineering Research Center of Intelligent Vision and Imaging. This work was also supported in part by computational resources provided by Fcloud CO., LTD.

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

# UniHM: Unified Dexterous Hand Manipulation with Vision Language Model
# Appendix

## A  USE OF LLMs

We used an GPT to polish the experiments and introduction sections. We also use the Nano to generate the icon in Figure 1.

## B  IMPLEMENTATION DETAILS

### B.1  RETARGETING

**Hand Pose Retargeting.** The DexYCB (Chao et al., 2021) and OakInk (Yang et al., 2022) datasets provide MANO pose sequences (Loper et al., 2023), object point clouds, and associated object trajectories. We employ Dex-Retargeting (Qin et al., 2023) to map MANO joint configurations to multiple robotic hands via a differentiable inverse-kinematics objective with joint limits and device-specific kinematic calibration. Given a MANO pose $q_{\mathrm{MANO}}(t)$, the device configuration is:

$$q_{\mathrm{dev}}(t) = \underset{q \in [q_{\min}, q_{\max}]}{\arg\min} \sum_k w_k \left\| f_k(q) - f_k(\phi_{\mathrm{dev}}(q_{\mathrm{MANO}}(t))) \right\|_2^2 + \lambda \left\| q - q_{\mathrm{prev}} \right\|_2^2, \quad \text{(B1)}$$

where $f_k$ denote forward-kinematics constraints on fingertip/phalange keypoints, $\phi_{\mathrm{dev}}$ aligns MANO and device frames and scales link lengths, and $q_{\mathrm{prev}}$ is the previous solution for temporal smoothness. We generate retargeted sequences for Allegro, Shadow, Schunk, LEAP, and Ability hands, and for the Panda gripper (mapping the thumb-index aperture to a scalar opening width via linear scaling with clamping).

**Target Trajectory Generation.** Let $M_{\mathrm{ext}} \in SE(3)$ denote the camera extrinsics that map world coordinates to the camera frame, and let $\mathcal{T}_{\mathrm{obj}}^{\mathrm{cam}}(t) \in SE(3)$ be the object pose in the camera frame at time $t$. The object target trajectory in the world frame used during training is

$$\mathcal{T}_{\mathrm{tar}}(t) = M_{\mathrm{ext}}^{-1} \mathcal{T}_{\mathrm{obj}}^{\mathrm{cam}}(t). \quad \text{(B2)}$$

We time-align $\mathcal{T}_{\mathrm{tar}}(t)$ with the retargeted hand poses using the dataset timestamps.

### B.2  UNIFIED HAND TOKENIZER

**Unified Codebook Training.** We adopt a shared VQ-VAE codebook $\mathcal{Z} = \{\mathbf{e}_k\}_{k=1}^K$ initialized with Shadow Hand due to its data availability and wildly used. For any hand type $h$, denote its encoder and decoder by $E_h$ and $D_h$. Given an input sequence chunk $\mathbf{x}^{(h)}$, the encoder output, quantization, and reconstruction follow the main text:

$$\mathbf{z}_e^{(h)} = E_h(\mathbf{x}^{(h)}), \quad \text{(B3)}$$

$$c = Q(\mathbf{z}_e^{(h)}) = \arg \min_{k \in [K]} \left\| \mathbf{z}_e^{(h)} - \mathbf{e}_k \right\|_2^2, \quad \text{(B4)}$$

$$\mathbf{z}_q^{(h)} = \mathbf{e}_c, \quad \text{(B5)}$$

$$\hat{\mathbf{x}}^{(h)} = D_h(\mathbf{z}_q^{(h)}) = D_h(\mathbf{e}_c), \quad \text{(B6)}$$

where $Q(\cdot)$ performs a nearest neighbor lookup in the codebook, finding the code vector with the minimum squared Euclidean distance. We train with the standard reconstruction and VQ losses:

$$\mathcal{L}_{\mathrm{rec}} = \left\| \mathbf{x}^{(h)} - D_h\left(\mathbf{z}_q^{(h)}\right) \right\|_2^2, \quad \text{(B7)}$$

$$\mathcal{L}_{\mathrm{vq}} = \left\| \mathrm{sg}\left[\mathbf{z}_e^{(h)}\right] - \mathbf{z}_q^{(h)} \right\|_2^2 + \beta \left\| \mathbf{z}_e^{(h)} - \mathrm{sg}\left[\mathbf{z}_q^{(h)}\right] \right\|_2^2, \quad \text{(B8)}$$

where $\mathrm{sg}[\cdot]$ is the stop-gradient operator and $\beta > 0$ is the commitment weight.

To avoid codebook collapse with a purely statistical procedure, we maintain a "cold-code" table based on per-epoch usage counts. Let $n_k^{(t)}$ be the usage of code $k$ during epoch $t$, computed from token indices $\{c_b^{(t)}\}_{b=1}^{B^{(t)}}$:

$$n_k^{(t)} = \sum_{b=1}^{B^{(t)}} \mathbb{1}\left[ c_b^{(t)} = k \right], \tag{B9}$$

where $\mathbb{1}(\cdot)$ denotes the indicator function that evaluates to 1 if the condition inside its brackets is true, and 0 otherwise. Define the cold set at epoch $t$ by a count threshold $\tau_c$:

$$\mathcal{S}^{(t)} = \left\{ k \in [K] \mid n_k^{(t)} < \tau_c \right\}. \tag{B10}$$

At scheduled refresh epochs $t \in \mathcal{E}_{\text{reset}}$, we selectively update only the codes belonging to the cold set. Collect a buffer $\mathcal{B}^{(t)}$ of encoder outputs since the previous refresh and run K-Means with $R = \left| \mathcal{S}^{(t)} \right|$ to obtain centroids $\{\boldsymbol{\mu}_r\}_{r=1}^R$. We directly replace the cold codes with the newly computed centroids. With an injective assignment $\pi : \mathcal{S}^{(t)} \to \{1, \dots, R\}$:

$$\mathbf{e}_k \leftarrow \boldsymbol{\mu}_{\pi(k)} \quad \text{for } k \in \mathcal{S}^{(t)}, \qquad \mathbf{e}_k \leftarrow \mathbf{e}_k \quad \text{for } k \notin \mathcal{S}^{(t)}. \tag{B11}$$

Thus, only the codes in the cold-code table are refreshed at those epochs, while frequently used codes remain unchanged. The objective for this stage is:

$$\mathcal{L}_{\text{loss}} = \mathcal{L}_{\text{rec}} + \mathcal{L}_{\text{vq}}. \tag{B12}$$

**Multi-Encoder and Decoder Training.** To integrate a new hand morphology, we first align the new encoder to the reference encoder via retargeted pairs. Let $E_{\text{ref}}$ be the reference (Shadow Hand) encoder and $(\mathbf{x}_{\text{new}}, \mathbf{x}_{\text{ref}})$ be a retargeted pair:

$$\mathcal{L}_{\text{distill}} = \| E_{\text{new}}(\mathbf{x}_{\text{new}}) - E_{\text{ref}}(\mathbf{x}_{\text{ref}}) \|_2^2. \tag{B13}$$

After encoder alignment, we train the new encoder–decoder pair within the shared VQ-VAE using the same reconstruction and VQ losses:

$$\mathcal{L}_{\text{total}}^{(h)} = \mathcal{L}_{\text{rec}} + \mathcal{L}_{\text{vq}} + \lambda_{\text{distill}} \mathcal{L}_{\text{distill}} \tag{B14}$$

, where $\lambda_{\text{distill}} \geq 0$ controls the contribution of the distillation term during the encoder alignment stage.

**Training Details.** We train a unified codebook with capacity $K = 8192$ on DexYCB sequences. Since the inputs are 1D pose and short-motion signals, we instantiate encoders/decoders with either MLP or 1D convolutional backbones and ablate both choices, as shown in Table B1. When training the reference encoder, we set the $\beta = 0.25$ and train the network with a learning rate $1e-4$. We also update the codebook at epochs $50, 100, 150, \dots$. For the multi-encoder and multi-decoder training, the cold-code threshold $\tau_c$ is set to 1, and the distillation weight $\lambda_{\text{distill}}$ is set to 0.1.

Table B1: A comparison of the validation set performance for the MLP and 1D-Conv models.

|  |  | Allegro | Shadow | Schunk | LEAP | Ability | Panda Gripper | Overall |
|---|---|---|---|---|---|---|---|---|
| MAE | MLP | 0.0268 | 0.0450 | 0.0293 | 0.0348 | 0.0432 | 0.0221 | 0.0350 |
|  | 1D-Conv | **0.0216** | **0.0297** | **0.0221** | **0.0257** | **0.0327** | **0.0182** | **0.0256** |
| RMSE | MLP | 0.0545 | 0.0886 | 0.0656 | 0.0656 | 0.0913 | 0.0630 | 0.0736 |
|  | 1D-Conv | **0.0465** | **0.0654** | **0.0551** | **0.0531** | **0.0705** | **0.0519** | **0.0581** |

### B.3 PHYSICAL OPTIMIZATION

**Physical Model.** We expand the contact term used in the per-frame objective. Let $s_i(q_t)$ denote the $i$-th fingertip position (forward kinematics) in the world frame and $\mathcal{T}_{\text{tar}}(t)^{-1}$ be the object-frame transform at time $t$. Query the nearest neighbor $(\mathbf{p}_i, \mathbf{n}_i)$ on $\mathcal{P}_{\text{obj}}$ in the object frame, and define the signed point-to-plane distance:

$$d_i(q_t) = \mathbf{n}_i^{\text{T}} (\mathcal{T}_{\text{tar}}(t)^{-1} s_i(q_t) - \mathbf{p}_i). \tag{B15}$$

We adopt an asymmetric, smooth penalty $f(d)$ that is continuous and slope-matched at $d = 0$:

$$f(d) = \begin{cases} \frac{\alpha}{2} d^2, & d \geq 0 \text{ (outside)} \\ \frac{\alpha}{k^2} \left( e^{-kd} + kd - 1 \right), & d < 0 \text{ (inside)} \end{cases} \tag{B16}$$

At $d = 0$, $f$ is continuous and has matched first derivatives:

$$\lim_{d \to 0^\pm} f(d) = 0, \quad \lim_{d \to 0^+} f'(d) = 0, \quad \lim_{d \to 0^-} f'(d) = \frac{\alpha}{k}(1 - e^{-k \cdot 0}) = 0. \tag{B17}$$

Moreover, $f''(0^-) = f''(0^+) = \alpha e^{-k \cdot 0} \geq 0$, so $f$ is twice-differentiable and convex. Stacking per-fingertip residuals gives:

$$r_{\text{contact},i}(q_t) = \sqrt{2\lambda_c f(d_i(q_t))}, \quad \mathcal{E}_{\text{contact}}(q_t) = \tfrac{1}{2}\|r_{\text{contact}}(q_t)\|_2^2 = \lambda_c \sum_i f(d_i(q_t)), \tag{B18}$$

with $\lambda_c > 0$. For Gauss–Newton, we linearize the residuals at the current iterate. Let $x_i(q_t) = \mathcal{T}_{\text{tar}}(t)^{-1} s_i(q_t)$ and $J_{x,i}(q_t) = \partial x_i / \partial q_t$ (the kinematic Jacobian in the object frame). Treating $(\mathbf{p}_i, \mathbf{n}_i)$ as fixed within an inner iteration:

$$\frac{\partial d_i(q_t)}{\partial q_t} = \mathbf{n}_i^{\mathrm{T}} J_{x,i}(q_t), \quad \frac{\partial r_{\text{contact},i}}{\partial q_t} = \sqrt{2\lambda_c} \frac{f'(d_i(q_t))}{2\sqrt{f(d_i(q_t)) + \epsilon^2}} \mathbf{n}_i^{\mathrm{T}} J_{x,i}(q_t), \tag{B19}$$

which yields the contact Jacobian rows that form $J_t = \partial r_{\text{contact}}(q_t)/\partial q_t$ with a small $\epsilon > 0$.

**Generative Prior and Temporal Prior.** We keep the generative and temporal priors from the main text and unify their roles as dynamic smoothers from two information sources—semantic intent (generator) and history (velocity/acceleration). The generative prior anchors $q_t$ to $q_t^{\text{gen}}$:

$$\mathcal{E}_{\text{gen}}(q_t, q_t^{\text{gen}}) = \tfrac{1}{2}(q_t - q_t^{\text{gen}})^{\mathrm{T}} \mathbf{W}_{\text{gen}}(q_t - q_t^{\text{gen}}), \quad \mathbf{W}_{\text{gen}} \succ \mathbf{0}. \tag{B20}$$

The temporal prior penalizes first- and second-order differences:

$$\mathcal{E}_{\text{time}}(q_t, q_{t-1}^{\text{opt}}, q_{t-2}^{\text{opt}}) = \tfrac{1}{2}\|q_t - q_{t-1}^{\text{opt}}\|_{\mathbf{W}_{\text{vel}}}^2 + \tfrac{1}{2}\|(q_t - q_{t-1}^{\text{opt}}) - (q_{t-1}^{\text{opt}} - q_{t-2}^{\text{opt}})\|_{\mathbf{W}_{\text{acc}}}^2, \tag{B21}$$

where $\mathbf{W}_{\text{vel}} \succ \mathbf{0}$ and $\mathbf{W}_{\text{acc}} \succ \mathbf{0}$ (often $\mathbf{W}_{\text{vel}} = \lambda_{\text{vel}} I$, $\mathbf{W}_{\text{acc}} = \lambda_{\text{acc}} I$). Define the per-frame objective:

$$\mathcal{E}_t(q_t, q_t^{\text{gen}}, q_{t-1}^{\text{opt}}, q_{t-2}^{\text{opt}}) = \mathcal{E}_{\text{contact}}(q_t) + \mathcal{E}_{\text{gen}}(q_t, q_t^{\text{gen}}) + \mathcal{E}_{\text{time}}(q_t, q_{t-1}^{\text{opt}}, q_{t-2}^{\text{opt}}). \tag{B22}$$

Let $r_{\text{contact}}(q_t)$ denote the stacked contact residuals. Linearizing $r_{\text{contact}}(q_t + \Delta q_t) \approx r_{\text{contact}}(q_t) + J_t \Delta q_t$ and expanding the two quadratic priors around $q_t$ give:

$$\mathcal{E}_t(q_t + \Delta q_t) \approx \tfrac{1}{2}\|r_{\text{contact}}(q_t) + J_t \Delta q_t\|_2^2 + \tfrac{1}{2}\Delta q_t^{\mathrm{T}}(\mathbf{W}_{\text{gen}} + \mathbf{W}_{\text{vel}} + \mathbf{W}_{\text{acc}})\Delta q_t + \Delta q_t^{\mathrm{T}}\tilde{\mathbf{W}}, \tag{B23}$$

where:

$$\tilde{\mathbf{W}} \triangleq \mathbf{W}_{\text{gen}}(q_t - q_t^{\text{gen}}) + \mathbf{W}_{\text{vel}}(q_t - q_{t-1}^{\text{opt}}) + \mathbf{W}_{\text{acc}}((q_t - q_{t-1}^{\text{opt}}) - (q_{t-1}^{\text{opt}} - q_{t-2}^{\text{opt}})). \tag{B24}$$

Setting the gradient w.r.t. $\Delta q_t$ to zero and adding Levenberg–Marquardt damping $\lambda \geq 0$ yields the normal equations:

$$(J_t^{\mathrm{T}} J_t + \mathbf{W}_{\text{gen}} + \mathbf{W}_{\text{vel}} + \mathbf{W}_{\text{acc}} + \lambda I)\Delta q_t = -J_t^{\mathrm{T}} r_{\text{contact}}(q_t) - \tilde{\mathbf{W}}. \tag{B25}$$

For $t < 2$, we use boundary priors consistent with the main text, e.g., $q_{-2}^{\text{opt}} = q_{-1}^{\text{opt}} = q_0^{\text{gen}}$.

**Optimization Procedure.** Given the decoded grasping trajectory $\mathcal{Q}_{\text{gen}}$, the object point cloud $\mathcal{P}_{\text{obj}}$, and the target pose trajectory $\mathcal{T}_{\text{tar}}(t)$, we refine each frame by Gauss–Newton with LM damping while freezing nearest-neighbor correspondences inside each inner iteration. In our implementation, we set $\alpha = k = 1$ and use a small positive constant $\epsilon$ to handle cases where $f(d)$ approaches zero.

In practical environments, obtaining precise point clouds is often challenging. A significant advantage of employing the kernel function $f(d)$ is the inherent robustness it provides to the segmentation process.

As shown in Fig. B1, the kernel function is relatively flat in a neighborhood of $d = 0$, which implies that, even when the segmented point cloud is contaminated with noise, the resulting cost term does not deviate substantially from that constructed from the ground-truth point cloud, and thus the optimization still converges to the correct solution in the presence of noisy inputs. We further demonstrate the optimization results under noisy conditions (for a clearer illustration of the grasp point, we have omitted the remaining structure of the dexterous hand, plotting only the object point cloud and the positions of the hand's fingers), as shown in Fig. B2.

---

**Algorithm 1** Physical-guided dynamic refinement

---

**Require:** Generated sequence $\mathcal{Q}_{\text{gen}} = \{q_t^{\text{gen}}\}_{t=0}^{T}$, point cloud $\mathcal{P}_{\text{obj}}$, target poses $\{\mathcal{T}_{\text{tar}}(t)\}_{t=0}^{T}$, weights $\lambda_c$, $\mathbf{W}_{\text{gen}}$, $\mathbf{W}_{\text{vel}}$, $\mathbf{W}_{\text{acc}}$, damping $\lambda \geq 0$
**Ensure:** Refined sequence $\mathcal{Q}_{\text{opt}} = \{q_t^{\text{opt}}\}_{t=0}^{T}$
1: Set boundary priors $q_{-2}^{\text{opt}} = q_{-1}^{\text{opt}} = q_0^{\text{gen}}$
2: **for** $t = 0$ to $T$ **do**
3:     Initialize $q_t \leftarrow q_t^{\text{gen}}$
4:     **repeat**
5:         Transform fingertips $x_i(q_t) = \mathcal{T}_{\text{tar}}(t)^{-1} s_i(q_t)$
6:         For each fingertip $i$, find $(\mathbf{p}_i, \mathbf{n}_i) = \text{NN}(x_i(q_t), \mathcal{P}_{\text{obj}})$ and determine inside/outside sign
7:         Compute $d_i(q_t)$, $r_{\text{contact}}(q_t)$, and $J_t = \partial r_{\text{contact}}/\partial q_t$ with correspondences fixed
8:         Form $\tilde{\mathbf{W}}$ using $q_t$, $q_{t-1}^{\text{opt}}$, $q_{t-2}^{\text{opt}}$
9:         Solve $(J_t^{\text{T}} J_t + \mathbf{W}_{\text{gen}} + \mathbf{W}_{\text{vel}} + \mathbf{W}_{\text{acc}} + \lambda I)\Delta q_t = -J_t^{\text{T}} r_{\text{contact}}(q_t) - \tilde{\mathbf{W}}$
10:        Update: $q_t \leftarrow q_t + \Delta q_t$; adjust $\lambda$ by decrease/increase of $\mathcal{E}_t$
11:     **until** converged or max iters
12:     $q_t^{\text{opt}} \leftarrow q_t$
13: **end for**

---

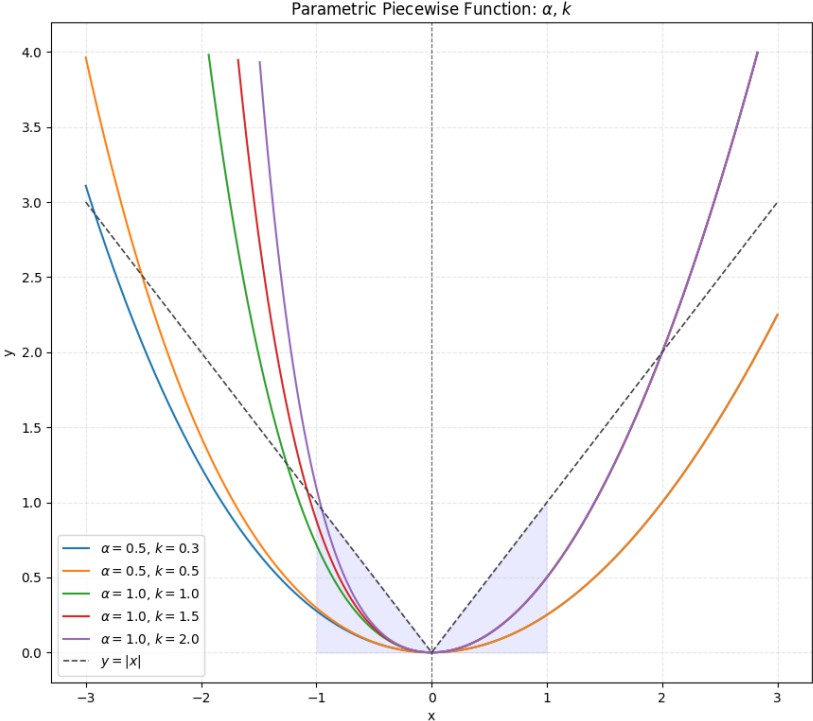

Figure B1: The function plots under varying values of $\alpha$ and $k$ are displayed. Note that $\alpha$ and $k$ control the curve behavior for $x > 0$ and $x < 0$, respectively. For comparison, the curve of $y = |x|$ is plotted using a dashed line.

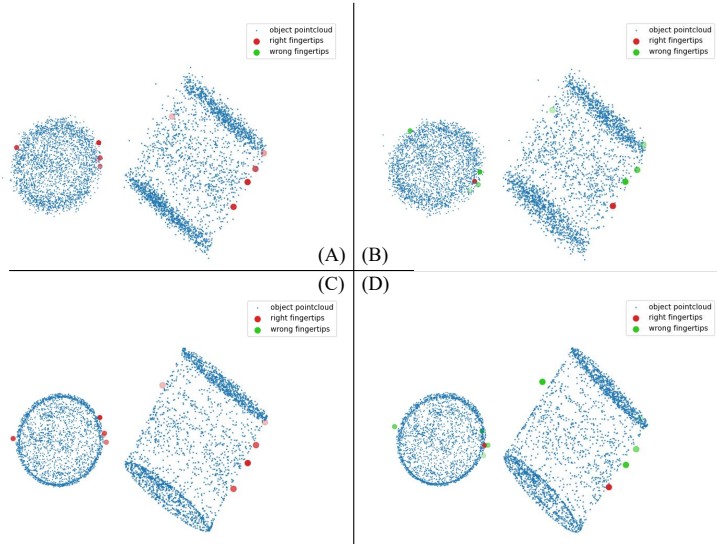

Figure B2: Optimization results when the point cloud includes noise. Here, the black points represent the object point cloud, and the red/green points denote the positions of the dexterous hand's fingertips. (A) Optimization using the $f(d)$ kernel, visualized on the noisy input. (B) Optimization using the Euclidean distance, visualized on the noisy input. (C) The $f(d)$ kernel optimization result projected onto the clean noise-free point cloud. (D) The Euclidean distance optimization result projected onto the noise-free point cloud.

## C  MODEL INSTRUCTION

### C.1  TEXT TOKENIZER

We employ a pretrained language model from the Qwen series, specifically Qwen3, as the core component for our language input pipeline. This choice is motivated by its strong performance across a wide range of natural language tasks and its robust, efficient tokenizer.

The Qwen3 tokenizer operates on a byte-level Byte-Pair Encoding algorithm (Sennrich et al., 2015). This method ensures that all possible input characters are representable, and it is particularly effective at handling diverse linguistic inputs, including code and technical terms, without resorting to unknown tokens. The tokenizer's vocabulary size is $V_{tok}$, and each input sequence $S$ is tokenized into a sequence of tokens $T = (t_1, t_2, \ldots, t_N)$, where $N$ is the sequence length. The tokenization process can be formally expressed as:

$$\text{tokenize}(S) \rightarrow T. \tag{C1}$$

This token sequence $T$ is then converted into a sequence of token embeddings, which serve as the input to the subsequent layers of our model. We utilize the tokenizer's built-in padding and truncation functionalities to handle variable-length sequences, ensuring a consistent input shape for the model.

### C.2  POINT CLOUD ENCODER

We use a PointNet-based architecture (Qi et al., 2017a;b) as our point cloud feature extractor, which is a key component of our model. The network directly consumes raw point cloud data, which is an unordered set of $n$ points, with each point represented by its $(x, y, z)$ coordinates. The PointNet architecture addresses the permutation invariance of point clouds by using a symmetric function to aggregate information from each point.

Specifically, the network first applies a shared multi-layer perceptron to each point individually to transform the input features. This is followed by a max-pooling layer, which acts as a symmetric function to aggregate the point-wise features into a global feature vector. This process can be concisely described as:

$$f(X) = \max_{i=1,\ldots,n} \{h(x_i)\}, \tag{C2}$$

where $X = \{x_1, \ldots, x_n\}$ is the input point cloud, $h$ represents the shared MLP, and $f(X)$ is the resulting global feature vector. The network is configured to output several feature vectors, which are then concatenated and fed into the subsequent main model for further processing.

### C.3 TRAJECTORY ENCODER

Our model takes the target trajectory $\mathcal{T}_{\text{tar}}$ as input. We encode this trajectory using a two-layer MLP to obtain a feature representation $\mathbf{z}(t)$. The MLP processes the pose $\mathbf{p}_{\text{tar}}(t)$ at each timestep $t$ and is defined as:

$$\mathbf{z}(t) = \sigma_2 \left(\mathbf{W}_2 \sigma_1 \left(\mathbf{W}_1 \mathbf{p}_{\text{tar}}(t) + \mathbf{b}_1\right) + \mathbf{b}_2\right), \tag{C3}$$

where $\sigma_1$ and $\sigma_2$ are non-linear activation functions. The hidden layer dimension is set to 512 during training.

### C.4 MASKED TRAINING

During training, we progressively mask the ground truth hand poses to encourage the model to learn the temporal relationships within a sequence. This approach, which has been shown to be highly effective in fields like language modeling (e.g., BERT (Liu et al., 2019)) and computer vision (e.g., DINOv2 (Oquab et al., 2023)), allows the model to generate hand poses from a given context.

Specifically, we feed the ground truth trajectory into the VQ-VAE encoder and then use a mask to conceal the corresponding hand poses. The masked hand pose tokens are replaced with a single, learnable token. This masking process can be formalized as:

$$\mathcal{Q}_{\text{masked}} = M \odot E(\mathcal{Q}) + (1 - M) \odot \mathcal{T}_{\text{mask}}, \tag{C4}$$

where $\mathcal{Q}$ is the ground truth hand pose sequence, $E$ is the VQ-VAE encoder, $M$ is a binary mask, and $\mathcal{T}_{\text{mask}}$ is the learnable token.

We implement this masking strategy with a progressive curriculum across training epochs. For the first 20% of epochs, we do not mask any hand poses. From 20% to 80% of the epochs, we linearly increase the masking ratio until it reaches 100%. Finally, during the last 20% of epochs, all hand poses are consistently masked, forcing the network to perform full generation. We train the model for 100 epochs using the AdamW optimizer with a learning rate of 1e-4.

### C.5 TARGET TRAJECTORY PLANER

We use CLiPort to plan the target trajectory $\mathcal{T}_{tar}$. Specifically, we take RGB-D and instructions as input data. Then, based on GenH2R (Wang et al., 2024), we generate a smooth trajectory in the output space, which is then encoded using the aforementioned trajectory encoder and fed into the model.

## D ADDITIONAL VISUALIZATION

We employ Sapien (Xiang et al., 2020) as a visualization engine for observing and validating the generated results. We also present some additional visualizations here.

## E REAL-WORLD SETUP

The experimental setup utilizes a Franka manipulator arm equipped with Panda Hand, an XHand dexterous hand, and Inspire Hand in Fig.E1. The 7-degree-of-freedom (DoF) Franka provides a

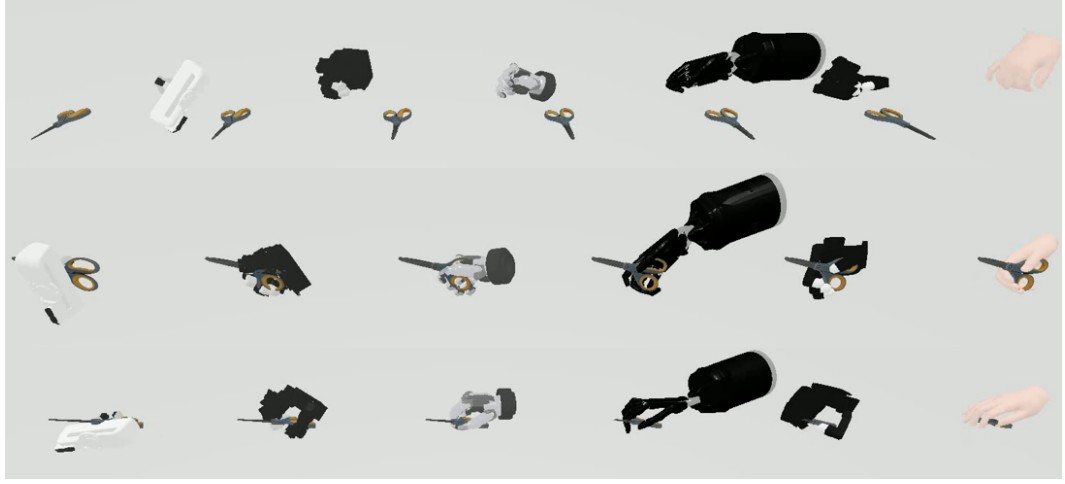

Figure D1: The visualization results of our grasping sequence in Sapien. More real world examples could be found in the video.

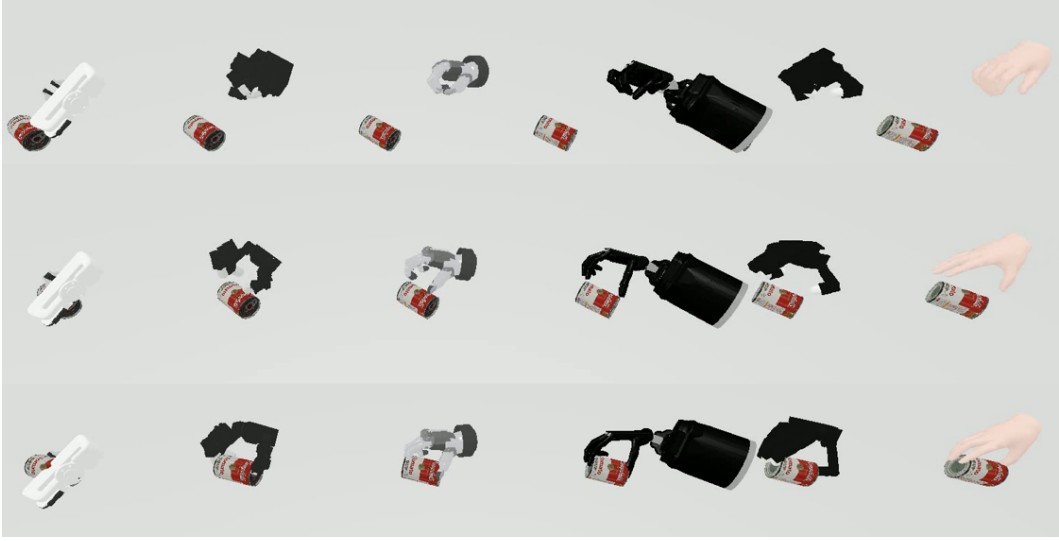

Figure D2: The visualization results of our grasping sequence in Sapien. More real world examples could be found in the video.

workspace comparable to that of a human arm. The 2-DOF Panda Hand, 12-DOF XHand, and 6-DOF Inspire Hand, with overall dimensions similar to a human hand, feature a proportionally elongated pinky finger. This specific design requires an adaptive scaling heuristic to ensure natural and fluid motion.

We employ a Zed camera to acquire RGB-D observations, apply PointSAM(You & Wu, 2025; Wang et al., 2025a; Wu et al., 2025) to segment the target point cloud, and subsequently leverage Snowflak-eNet (Xiang et al., 2023; 2021) to reconstruct a complete point cloud that serves as input to our model.

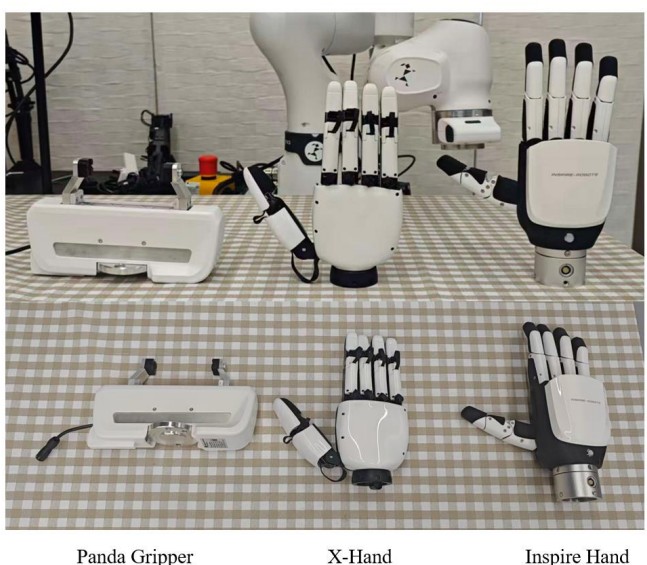

Panda Gripper        X-Hand        Inspire Hand

Figure E1: Real-World Cross-embodiment Set up

## F    EVALUATION METRIC

**MPJPE (Mean Per-Joint Position Error).** For each sequence, we compute the Euclidean error of every hand joint in every frame, Then average over joints and time:

$$\text{MPJPE}_i = \frac{1}{T_i J} \sum_{t=1}^{T_i} \sum_{j=1}^{J} \left\| p_{t,j}^{(i)} - g_{t,j}^{(i)} \right\|_2. \tag{F1}$$

The final MPJPE reported

$$\text{MPJPE} = \frac{1}{N} \sum_{i=1}^{N} \text{MPJPE}_i. \tag{F2}$$

where $N$ is the number of test sequences. **FPL (Final Position Location Error)** FPL measures how close the final hand placement is to the ground truth. Let $c_T^{(i)}$ and $\hat{c}_T^{(i)}$ be the 3D positions of the hand root / palm center at the last frame $T_i$ of sequence $i$:

$$\text{FPL}_i = \left\| \hat{c}_{T_i}^{(i)} - c_{T_i}^{(i)} \right\|_2, \qquad \text{FPL} = \frac{1}{N} \sum_i \text{FPL}_i. \tag{F3}$$

**FOL (Final Orientation Location Error).** FOL measures the orientation error of the hand at the final frame. Let $R_{T_i}^{(i)}$ and $\hat{R}_{T_i}^{(i)}$ be the ground-truth and predicted rotation matrices of the hand root; the orientation error is

$$\theta_i = \arccos\left( \frac{\text{trace}\left( (R_{T_i}^{(i)})^\top \hat{R}_{T_i}^{(i)} \right) - 1}{2} \right), \tag{F4}$$

converted to degrees. We report

$$\text{FOL} = \frac{1}{N} \sum_i \theta_i. \tag{F5}$$

**Feature extractor.** We first embed every hand-object interaction sequence $x$ (either real or generated) using a pretrained motion encoder $f(\cdot)$ that consumes the full temporal joint trajectory (and object pose) and outputs a fixed-dimensional feature vector $z = f(x) \in \mathbb{R}^d$. We use the same encoder for all methods and compute all distributional metrics in this learned feature space.

**FID (Fréchet Inception Distance).** Let $\{z_k^{\text{real}}\}_{k=1}^{M_r}$ be the features of real test sequences and $\{z_k^{\text{gen}}\}_{k=1}^{M_g}$ the features of generated sequences. We estimate the empirical means and covariances:

$$\mu_r = \frac{1}{M_r} \sum_k z_k^{\text{real}}, \quad \Sigma_r = \text{Cov}(\{z_k^{\text{real}}\}); \tag{F6}$$

$$\mu_g = \frac{1}{M_g} \sum_k z_k^{\text{gen}}, \quad \Sigma_g = \text{Cov}(\{z_k^{\text{gen}}\}), \tag{F7}$$

and compute

$$\text{FID} = \|\mu_r - \mu_g\|_2^2 + \text{Tr}\big(\Sigma_r + \Sigma_g - 2(\Sigma_r \Sigma_g)^{1/2}\big). \tag{F8}$$

**Diversity.** Diversity measures how varied the generated sequences are in the same feature space. Given the set of features $\{z_k\}_{k=1}^M$ for a method, we compute

$$\text{Div} = \frac{2}{M(M-1)} \sum_{1 \leq a < b \leq M} \|z_a - z_b\|_2. \tag{F9}$$

