# OpenReview forum: "UniHM: Unified Dexterous Hand Manipulation with Vision Language Model"
_ICLR.cc/2026/Conference — ICLR 2026 Poster_

### Official Review · Reviewer_uKvV · 2025-10-31

**Soundness:** 2
**Presentation:** 3
**Contribution:** 3
**Rating:** 6
**Confidence:** 4

**Summary:**

This paper explores the use of multimodal large models for cross-embodiment dexterous manipulation. The main contributions are as follows:
1. Constructed a dexterous manipulation dataset encompassing multiple heterogeneous dexterous hands, based on human manipulation datasets.
2. Proposed a language-guided, multimodal large model-based framework for cross-embodiment dexterous manipulation.

**Strengths:**

1.	The proposed dataset features highly aligned grasp poses with semantic annotations for cross-embodiment manipulation, which is quite a contribution to the community.
2.	Proposes and validates a viable methodology for applying Multimodal Large Language Models (MLLMs) to the challenging problem of high-DoF dexterous manipulation generation.
3.	Achieving promising results in real-world deployment on high-DoF dexterous embodiments using only human demonstration videos, providing a positive signal for addressing the issue of expensive data collection.

**Weaknesses:**

1.	The selected tasks are relatively simple. On one hand, many demonstrated actions (e.g., opening a drawer) can be performed by simpler, lower-cost two-fingered grippers, failing to highlight the advantage of dexterous hands. On the other hand, the tasks designed for dexterous hands (e.g., picking and placing objects) involve overly simplistic semantics, lacking part-level manipulation or interactions with clear subsequent intent (e.g., "handing the scissor handles to the user" or "grasping the teapot lid", which are feasible in OakInk). We expect to see complex manipulation sequences that are hard to do for non-dexterous hands and feature richer semantic hierarchies.
2.	The evaluations in simulation are primarily analytical metrics. I believe success rates for the manipulations are essential.

**Questions:**

1.	In real-world deployment in open-world scenarios, high-fidelity 3D assets of the objects being manipulated are unavailable. Could this affect the performance of the physics-guided dynamic refinement?
2.	The experiments just include comparisons with several human motion generation models. What about the performance of language-guided Hand-Object Interaction models [1] or language-guided grasp models [2, 3] (which can extend to the setting of this paper) on the current dataset.

[1] Cha, Junuk, et al. "Text2hoi: Text-guided 3d motion generation for hand-object interaction." Proceedings of the IEEE/CVF Conference on Computer Vision and Pattern Recognition. 2024.

[2] Wei, Yi-Lin, et al. "Grasp as you say: Language-guided dexterous grasp generation." Advances in Neural Information Processing Systems 37 (2024): 46881-46907.

[3] Zhong Y, Huang X, Li R, et al. Dexgraspvla: A vision-language-action framework towards general dexterous grasping[J]. arXiv preprint arXiv:2502.20900, 2025.

---

> ### Author Response · Authors · 2025-11-21
>
> We sincerely appreciate Reviewer uKvV’s helpful comments. Below, we provide a point-by-point response to each of your remarks.
>
> > Q1: The selected tasks are relatively simple. On one hand, many demonstrated actions (e.g., opening a drawer) can be performed by simpler, lower-cost two-fingered grippers, failing to highlight the advantage of dexterous hands. On the other hand, the tasks designed for dexterous hands (e.g., picking and placing objects) involve overly simplistic semantics, lacking part-level manipulation or interactions with clear subsequent intent (e.g., "handing the scissor handles to the user" or "grasping the teapot lid", which are feasible in OakInk). We expect to see complex manipulation sequences that are hard to do for non-dexterous hands and feature richer semantic hierarchies.
>
> A1: Thank you for your suggestion. We have added real-world experiments in the supplementary video `complex_manipulation.mp4`, mainly focusing on **open-vocabulary semantics** and **part-level manipulation with clear subsequent intent**, such as "I am hungry, could you put some food on the white plate for me?","handing the scissor handles to the user","grasping the teapot lid".
>
> > Q2: The evaluations in simulation are primarily analytical metrics. I believe success rates for the manipulations are essential.
>
> A2: Thanks for your suggestion. We have selected a large number of complex grasping tasks in the simulator, and our experimental results are as follows.
> | Method      | DexYCB-Seen | OAKINK-Seen | DexYCB-Unseen | OAKINK-Unseen |
> |------------|-------------|-------------|---------------|---------------|
> | TM2T       | 6.4%        | 4.1%        | 5.0%          | 3.6%          |
> | MDM        | 8.5%        | 3.6%        | 7.2%          | 3.3%          |
> | FlowMDM    | 17.1%       | 14.5%       | 12.4%         | 9.8%          |
> | MotionGPT3 | 20.9%       | 17.9%       | 17.6%         | 15.3%         |
> | **Ours**   | **31.4%**   | **30.1%**   |**28.0%** | **25.1%** |

---

> > ### Author Response · Authors · 2025-11-21
> >
> > > Q3: In real-world deployment in open-world scenarios, high-fidelity 3D assets of the objects being manipulated are unavailable. Could this affect the performance of the physics-guided dynamic refinement?
> >
> >
> > A3: Yes, imprecise point cloud segmentation can indeed affect the accuracy of the algorithm. However, our optimization procedure exhibits robustness to this issue.
> >
> > Specifically, we employ a robust kernel function (Eq. B.16) to mitigate the detrimental influence of points arising from inaccurate segmentations. As shown in Fig. B1 of the appendix, the kernel function is relatively flat in a neighborhood of  $d=0$. This implies that, even when the segmented point cloud is contaminated with noise, the resulting cost term does not deviate substantially from that constructed from the ground-truth point cloud, and thus the optimization still converges to the correct solution in the presence of noisy inputs. We provide a detailed explanation (Lines 855–863), together with a toy example and supplementary visualizations illustrating the behavior of the kernel over a range of $\alpha$ and $k$ values, in Appendix B.3 (Fig. B1 and Fig. B2, Lines 855–943).
> >
> >
> > For real-world deployment, we further utilized a higher-fidelity RGB-D sensor and employed a point cloud completion algorithm to densify the segmented point clouds. Before, we didn't mention these details. In the revision, we have added them in the Appendix E(Lines 1084-1087) marked with blue color.
> >
> > > Q4:The experiments just include comparisons with several human motion generation models. What about the performance of language-guided Hand-Object Interaction models "Text2hoi" or language-guided grasp models "Grasp as you say, Dexgraspvla" (which can extend to the setting of this paper) on the current dataset.
> >
> > A4: We have added the following comparison.
> >
> > For Text2HOI, the performance of it on our dataset DexYCB was evaluated as follows.
> > | Split   | Method                   | MPJPE↓        | FOL↓         | FPL↓         | FID↓         | Diversity →    |
> > |--------|--------------------------|---------------|--------------|--------------|--------------|----------------|
> > |-   | GT                       | -             | -            | -            | -            | 125.53         |
> > | Seen   | Text2HOI |80.02±0.45     |26.13±0.94    | 18.63±2.28   |44.67±1.95    |58.14±1.79     |
> > | Seen   | **Ours** | **61.40±1.93**| **23.14±0.65**| **12.15±0.24**| **31.24±1.02**| **39.62±0.66**     |
> > | Unseen | Text2HOI |84.71±1.39   |29.25±1.76    |19.25±0.96    |48.39±1.53    | 55.13±0.92     |
> > | Unseen | **Ours**| **63.56±2.08**| **27.29±0.43**| **13.06±0.43**| **41.03±1.65**| **42.70±1.19**     |
> >
> >
> > For Grasp as you say, it is a static pose generation algorithm, whereas our method generates dynamic motion sequences. Therefore, a fair comparison between the two is only possible in real-world experiments.
> > | Split  | Method                         | Grab | Pick&Place | Pull&Push | Open&Close |
> > |--------|--------------------------------|------|-----------|-----------|------------|
> > | Seen   | Grasp as you say           |50%  |40%        |45%         |45%          |
> > | Seen   | **Ours**                      | **65%** | **50%** | **60%**  | **55%**    |
> > | Unseen | Grasp as you say           |35%    |30%         |40%       | 45%         |
> > | Unseen | **Ours**                      | **60%** | **35%** | **55%**  | **45%**    |
> >
> >
> > DexGraspVLA requires a large amount of real-robot teleoperation data for training and fails to converge on the HOI dataset used in our paper. Therefore, a direct comparison with our method is not feasible.

---

> > > ### Comment · Reviewer_uKvV · 2025-11-26
> > > **Comment**
> > >
> > > Thanks for your response. My concerns have been solved, therefore I will keep my score to support the accept of this paper.

---

### Official Review · Reviewer_ZSGv · 2025-10-31

**Soundness:** 2
**Presentation:** 3
**Contribution:** 2
**Rating:** 2
**Confidence:** 3

**Summary:**

The paper proposes a unified pipeline for language-conditioned, sequential dexterous-hand manipulation. The method comprises: (i) a Unified Dexterous-Hand Tokenizer (VQ-VAE) that maps heterogeneous hand morphologies into a shared discrete codebook and decodes back to hand-specific joint trajectories; (ii) a VLM-based generator that, given RGB-D input, an instruction, object point clouds, and tokenized history, autoregressively produces manipulation tokens; and (iii) a physics-guided dynamic refinement that enforces contact and temporal smoothness while adhering to the generator’s intent, yielding physically feasible, executable trajectories. Experiments on DexYCB and OakInk show consistent improvements over strong motion-generation baselines (TM2T, MDM, FlowMDM, MotionGPT3) and higher real-robot success rates; ablations demonstrate each module’s contribution.

**Strengths:**

The paper proposes a strong framework that incorporates a cross-dexterous-hand representation, language-conditioned sequence generation, and a physics-guided dynamic trajectory refinement module.

The authors perform extensive experiments across multiple datasets, covering both seen and unseen settings, with comprehensive evaluations and ablations.

**Weaknesses:**

Tokenizer evaluation limited to a single hand. Although a unified dexterous-hand tokenizer is proposed, both the HOI and real-world experiments appear to use only one hand type. A broader evaluation across multiple robot hands would more convincingly validate the tokenizer’s generality and cross-hand transfer.

Underspecified sequence-generation metrics. The paper introduces a manipulation sequence generator, but the evaluation protocol for sequences is insufficiently detailed. Please clearly define the quantitative metrics and how they are computed.

Missing text–motion alignment evaluation. Given the language-conditioned setup, include explicit Text–Motion Alignment assessments is necessary. Please provide both quantitative measures and qualitative analyses (e.g., human judgments of instruction adherence).

Minor issues.

Line 075: “Language -guided” → “Language-guided”.

Line 115: missing space before “introduces”.

Line 377: only five metrics are listed

**Questions:**

See weaknesses.

---

> ### Author Response · Authors · 2025-11-21
>
> We sincerely thank Reviewer ZSGv for the valuable comments and questions. Below, we provide point-by-point clarifications addressing the issues you raised.
>
> > Q1: Tokenizer evaluation limited to a single hand. Although a unified dexterous-hand tokenizer is proposed, both the HOI and real-world experiments appear to use only one hand type. A broader evaluation across multiple robot hands would more convincingly validate the tokenizer’s generality and cross-hand transfer.
>
> A1: We would like to clarify that actually we used **5 types of hand in HOI** and **2 types of hands in real-world**. Specifically, we consider the Panda hand (a two-finger, low-degree-of-freedom gripper) and four additional dexterous hands (multiple fingers, high degrees of freedom). The results for all five dexterous hands are reported in the appendix (Lines 1030-1073). To best showcase our cross-embodiment capability in real world, we focus on two representative embodiments: a low-DoF two-finger gripper and a high-DoF five-finger dexterous hand. Real-world experiments with these embodiments demonstrate the strong performance of our model.
>
> Furthermore, to more thoroughly validate the cross-embodiment generalization of our approach, we also conduct real-world experiments on **a new high-DoF dexterous hand**. The **demonstration video** `cross_embodiment.mp4` of the corresponding experiment has been added to the supplementary materials. The real-world experiment demonstrate that our method **generalizes effectively across different embodiments, including Panda Gripper, X-hand, and Inspire Hand**.
>
> > Q2: Underspecified sequence-generation metrics. The paper introduces a manipulation sequence generator, but the evaluation protocol for sequences is insufficiently detailed. Please clearly define the quantitative metrics and how they are computed.
>
> A2: Thanks for your suggestion. As we mentioned in our paper (Lines 408-422), these metrics are widely used in the area of hand manipulation. For your information, we also added the detailed explanation for these metrics into our Appendix marked with blue color (Lines 1111-1157).
>
>
> > Q3: Missing text–motion alignment evaluation. Given the language-conditioned setup, include explicit Text–Motion Alignment assessments is necessary. Please provide both quantitative measures and qualitative analyses (e.g., human judgments of instruction adherence).
>
> A3: For quantitative evaluation, we report FID as the text–motion alignment metric, which is also widely used in classic motion generation works such as MDM [R4], MotionGPT [R5], and Text2HOI [R6].
> For qualitative analysis, we additionally conduct a user study based on human judgments of instruction adherence, as detailed below. For each comparison, we perform 100 trials.
>
> | Compare      | Prefer UniHM | Prefer the Other | no Prefer |
> |------------|-------------|-------------|---------------|
> | UniHM vs. TM2T       | 95%        | 1%        | 4%          |
> | UniHM vs. MDM        | 91%        | 3%        | 6%          |
> | UniHM vs. FlowMDM    | 83%        | 9%        | 8%          |
> | UniHM vs. MotionGPT3 | 71%        | 22%       | 7%          |
> | UniHM vs. GT         | 34%        | 61%       | 5%          |
>
> [R4] Human Motion Diffusion Model
> [R5] MotionGPT: Human Motion as a Foreign Language
> [R6] Text2HOI: Text-guided 3D Motion Generation for Hand-Object Interaction
>
> > Q4: Line 075: “Language -guided” → “Language-guided”.
>
> A4: Thanks for your suggestions. We have corrected this typo and noted it in the manuscript (Line 53).
>
> > Q5: Line 115: missing space before “introduces”.
>
> A5: Thanks for your suggestions. We have corrected this typo and noted it in the manuscript (Line 115).
>
> > Q6: Line 377: only five metrics are listed
>
> A6: Thanks for your suggestions. We have corrected this typo and noted it in the manuscript (Line 413).

---

### Official Review · Reviewer_7Tpz · 2025-10-31

**Soundness:** 3
**Presentation:** 3
**Contribution:** 3
**Rating:** 8
**Confidence:** 2

**Summary:**

In this work, the authors propose a novel framework for manipulation that generalizes across different dexterous-hand morphologies. Furthermore, the proposed method can generalize over unseen objects. The method consists of three main stages: first, the motion is tokenized in a morphology-independent way. Second, a token sequence is generated based on combined text, perception, and token history. Finally, the motion is generated by using physics-aware decoding. The authors report results that show that their method is superior to baselines on existing datasets, as well as in real-world evaluations.

**Strengths:**

The paper is considering a relevant problem of generalization to different hand morphologies. The authors proposed a comprehensive framework that shows good results across multiple metrics and in real-world robot setups. They successfully manage to combine and benefit from existing models, such as PointSam and CLIPort, and integrate them into their framework. Overall, the approach is well-defined, and evaluation supports the claims.

**Weaknesses:**

The main weakness of the approach is the complexity of the method. Currently, the approach consists of multiple parts and many different pre-trained models that need to be finetuned.

The authors do not provide code. It is unclear which simulation environment they have used.

The authors mention some terms without providing sufficient explanation in their context or a reference to related work. Such terms are MANO poses (line 153), vector-quantization operator (line 187), knowledge distillation (line 203).

Smaller writing comments and typos:
- In the introduction, the authors listed 4 core capabilities, and 4 contributions. However, they almost identically, and their repetition is not well motivated. Therefore, the authors should either differentiate them better, or combine them in order to improve legibility.
- Line 045: No space before AffordDexGrasp
- Line 272: No space between sentences (generation.A practical)
- Table 1, 2, and 4: The arrow pointing to right next to Diversity is unintuitive, and its meaning might confuse readers, who did not read the evaluations section in detail.

**Questions:**

Have you tried different VLMs?

Have you tried others pretrained models instead of PointSam and CLIPort?

Have you tried different capacities of the Codebook?

Which simulation engine have you used? In the Appendix D it is mentioned that Sapien is used for visualization.

---

> ### Author Response · Authors · 2025-11-21
>
> We thank Reviewer 7Tpz for the detailed feedback and the opportunity to clarify our work. We provide comprehensive responses to these concerns below.
>
> > Q1:The main weakness of the approach is the complexity of the method. Currently, the approach consists of multiple parts and many different pre-trained models that need to be finetuned.
>
> A1: Yes, our framework partially relies on the pre-trained and zero-shot capabilities of the VLM. However, this design enables our model to achieve powerful performance **without** requiring expensive teleoperation data collection; we only need to train on existing HOI datasets.
>
> > Q2:The authors mention some terms without providing sufficient explanation in their context or a reference to related work. Such terms are MANO poses (line 153), vector-quantization operator (line 187), knowledge distillation (line 203).
>
> A2: MANO [R1] is a widely used parametric 3D hand model that represents the geometry and pose of an arbitrary hand using a set of parameters. MANO poses refer to the sequence representation of the hand in HOI datasets, i.e., the hand pose sequence along the interaction trajectory.
>
> The vector-quantization [R2] operator compresses high-dimensional data into a set of discrete codes. Formally, it is defined as a function $\mathcal{Q}(\mathbf{z})$ that takes a latent vector $\mathbf{z}$ as input and maps it to the nearest codebook vector in $E = \{e_1, ..., e_K\}$, i.e.,
> $Q(z) = \arg\min_{e_i \in E} \|z - e_i\|_2.$.
>
> Knowledge distillation [R3] is a training technique that transfers the “knowledge” learned by a large, high-capacity model (the teacher model) to a smaller, more efficient model (the student model). In our work, however, we leverage a high–information-density model (an encoder trained on Shadow Hand) to transfer its hand representation to lower–information-density models (encoders for various dexterous-hand parameterizations), and use the distillation process to align the latent spaces of these different encoders.
>
> [R1] Embodied Hands: Modeling and Capturing Hands and Bodies Together
> [R2] Neural Discrete Representation Learning
> [R3] Distilling the Knowledge in a Neural Network
>
>
> > Q3:In the introduction, the authors listed 4 core capabilities, and 4 contributions. However, they almost identically, and their repetition is not well motivated. Therefore, the authors should either differentiate them better, or combine them in order to improve legibility.
>
> A3: Thanks for your suggestions. We have rewritten this section and distilled the original four core capabilities into two core capabilities. We have noted the corresponding sections in the manuscript(Lines 52-81).
>
>
>
> > Q4:Line 045: No space before AffordDexGrasp.
>
> A4:Thanks for your suggestions. We have corrected this typo and noted it in the manuscript (Line 45).
>
> > Q5:Line 272: No space between sentences (generation.A practical)
>
> A5: Thanks for your suggestions. We have corrected this typo and noted it in the manuscript (Line 277).

---

> > ### Author Response · Authors · 2025-11-21
> >
> > > Q6: Table 1, 2, and 4: The arrow pointing to right next to Diversity is unintuitive, and its meaning might confuse readers, who did not read the evaluations section in detail.
> >
> > A6: Thanks for your suggestions. We explained this in the table title. We have noted this modification in the manuscript (Line 325, Line 339, Line 433)
> >
> > > Q7: Have you tried different VLMs?
> >
> > A7: Yes. We experimented with VLMs of different parameter scales, including LLaVA-7B, Qwen-7B, Qwen-3B, and Qwen-0.6B, and found that Qwen-0.6B achieves the best performance. We attribute this to the limited size of current HOI datasets: larger VLMs are harder to optimize and often fail to converge reliably in such a low-data regime, leading to inferior results. This observation is consistent with prior motion generation work (e.g., MotionGPT), which also indicates that larger language backbones do not necessarily yield better performance when training data is scarce.
> >
> >
> > > Q8: Have you tried others pretrained models instead of PointSam and CLIPort?
> >
> > A8: Yes. We also tried other models such as SAM3D, Open3DSAM, and UniSeg3D instead of PointSAM, and ReKap, RoboBrain instead of CLIPort. However, after comparing inference speed, computational cost, stability, and adaptability to our setting, we ultimately chose PointSAM and CLIPort for our final system.
> >
> > > Q9: Have you tried different capacities of the Codebook?
> >
> > A9: Yes. We have experimented with codebooks of various sizes. During training, to avoid codebook collapse, we first perform a warm-up stage where we periodically reset the codebook using K-Means. After the warm-up, we monitor the usage frequency of each code and periodically replace cold codes using K-Means-based updates. We evaluated codebook sizes of 512, 1024, 2048, 4096, 8192, 16384, and 32768, and found that a size of 8192 provides the best trade-off between computational cost and codebook utilization.
> >
> >
> >
> > > Q10: Which simulation engine have you used? In the Appendix D it is mentioned that is used for visualization.
> >
> > A10: Yes, we choose Sapien as our simulation in our work. Sapien is a powerful Simulation for Physical optimization. Sapien is a widely adopted and prominent simulator within the field of embodied artificial intelligence. It offers a significant advantage, particularly in the high-fidelity simulation of articulated objects and kinematic chains, making it a robust platform for evaluating complex robotic manipulation and interaction tasks.

---

### Official Review · Reviewer_s9Mv · 2025-11-02

**Soundness:** 3
**Presentation:** 2
**Contribution:** 3
**Rating:** 6
**Confidence:** 4

**Summary:**

The paper presents a VLM approach for unified dexterous manipulation. Their approach consists of multiple components. First, a unified tokenizer is learned for multiple hand morphologies using a VQ-VAE codebook. The tokenizer is learned in such a way that all morphologies share the same codebook but different encoder and decoders. For the VLM training, qwen 0.6B is used as base modell. The VLM is trained to ouptut the hand configurations given the object target trajectory and the point cloud of the object. For inference, the point cloud is inferred using PointSAM and the target trajectory using CLIPort. Finally, the hand poses are refined via optimization using a cost function that aligns contact normals with surface normals of the point cloud as well as produces smooth motions. The approach is tested on several benchmark datasets as well as real robot tasks.

**Strengths:**

- First VLM for unified (multi-embodiment) dexterous manipulation
- The VLM can be trained purely from human data
- Performance seems to be competitive
- Real robot experiments are convincing

**Weaknesses:**

- Some sections are unclear and missing detail (see questions)
- The paper would benefit from further ablations. E.g. the different parts of the cost function. Or the benefits of using a unified latent space. Here, the (maybe naive) alternative would be to learn everything in a single space (e.g. MANO) and retarget the output of the VLM afterwards. Such a comparison would be insightful.
- There is no further information on the costs of the optimization of the grasp. How long does it take? Is it real-time capable?

However, I think the approach is interesting and shows a promising performance. The strengths do outweight the weaknesses.

**Questions:**

- I got confused in section 3.3 as its unclear how the mathematical objects in Eq. 7 and 8 look like. Could you please add more information here and clarify what T_tar and P_obj is (also formally, what is the dimensionality)?
- Is the target trajectory the 6D pose trajectory of an object or the 3D positions of a keypoint?

---

> ### Author Response · Authors · 2025-11-21
>
> We sincerely thank Reviewer s9Mv for the valuable comments and insightful questions. We address these points in detail below.
>
> > Q1: The paper would benefit from further ablations. E.g. the different parts of the cost function. Or the benefits of using a unified latent space. Here, the (maybe naive) alternative would be to learn everything in a single space (e.g. MANO) and retarget the output of the VLM afterwards. Such a comparison would be insightful.
>
> A1:Thanks for your suggestion, We provide the ablation study about "different parts of the cost function" and "using a unified latent space" below:
> **Ablation Study** about "different parts of the cost function" on DexYCB
> | Split  | Method              | MPJPE↓        | FOL↓          | FPL↓          | FID↓          | Diversity→     |
> |--------|---------------------|---------------|---------------|---------------|---------------|----------------|
> | -       | GT                  | -              |     -          |     -          |   -            |   125.53             |
> | Seen   | w/o $l_{\text{vq}}$     |121.65±14.46   |45.09±3.54   | 33.97±4.69              |83.65±4.89               | 225.75±23.17               |
> | Seen   | w/o $l_{\text{rec}}$      |113.07±3.53 | 48.37±2.03   | 36.25±1.75              |86.77±5.01               | 209.36±14.98               |
> | Seen   | w/o $l_{\text{gen}}$     | 63.07±2.25| 24.76±0.58     |14.46±0.67 | 32.44±1.06              |38.09±0.62                |
> | Seen   | w/o $l_{\text{time}}$    | 62.85±1.19| 24.07±0.93     |14.59±0.54  |32.5±0.28               |37.54±1.28                |
> | Seen   | **Ours**            | **61.40±1.93** | **23.14±0.65** | **12.15±0.24** | **31.24±1.02** | **39.62±0.66** |
> | Unseen | w/o $l_{\text{vq}}$     | 124.76±9.42   |43.97±2.09   | 34.98±2.26              | 88.98±3.74              | 197.65±23.07               |
> | Unseen | w/o $l_{\text{rec}}$      | 103.56±10.37  |49.14±0.73 | 35.41±3.39              | 92.15±6.08              | 223.76±5.09               |
> | Unseen | w/o $l_{\text{gen}}$     | 64.93±1.29    |27.96±0.58  | 14.32±0.59    |43.03±0.17               |40.03±0.37                |
> | Unseen | w/o $l_{\text{time}}$    | 64.02±1.17    |28.01±0.70  | 15.08±0.86     |44.28±0.49               |38.82±1.45                |
> | Unseen | **Ours**            | **63.56±2.08** | **27.29±0.43** | **13.06±0.43** | **41.03±1.65** | **42.70±1.27** |
>
>
> For $l_{\text{distill}}$ and $l_{\text{qpos}}$, removing these two loss terms makes the encoder distillation and VLM training completely fail, causing the model not to converge. Therefore, we are unable to report meaningful ablation metrics for them.
>
> **Ablation Study** about using MANO as a "unified latent space" on DexYCB
> | Split  | Method  | MPJPE↓        | FOL↓          | FPL↓          | FID↓          | Diversity →   |
> |--------|---------|---------------|---------------|---------------|---------------|---------------|
> |-       | GT      | -             | -             | -             | -             | 125.53        |
> | Seen   | MANO    |   67.19±2.35  | 25.46±0.93    | 15.32±0.40    | 37.06±1.98    | 37.19±0.63              |
> | Seen   | **Ours**| **61.40±1.93**| **23.14±0.65**| **12.15±0.24**| **31.24±1.02**| **39.62±0.66**|
> | Unseen | MANO    |   70.44±1.93  | 28.03±0.26    | 17.49±0.85    | 43.55±2.37    | 38.05±2.53    |
> | Unseen | **Ours**| **63.56±2.08**| **27.29±0.43**| **13.06±0.43**| **41.03±1.65**| **42.70±1.27**|
> > Q2: There is no further information on the costs of the optimization of the grasp. How long does it take? Is it real-time capable?
>
> A2: Yes, our optimization is real-time capable: for an entire manipulation sequence of 72 frames, we require only 0.63s for optimization, i.e., approximately 8.7ms per pose.
> We construct a k-d tree for the object point cloud to efficiently compute the closest points between the fingertips and the object, which accelerates convergence and facilitates early termination of the iterative process. In addition, we use the optimization result of the previous frame as the initial estimate for the subsequent frame and introduce a prior regularization term to alleviate issues such as divergence, and chattering, both of which further promote fast convergence and early stopping of the optimization.

---

> > ### Author Response · Authors · 2025-11-21
> >
> > > Q3: I got confused in section 3.3 as its unclear how the mathematical objects in Eq. 7 and 8 look like. Could you please add more information here and clarify what T_tar and P_obj is (also formally, what is the dimensionality)?
> >
> > A3: Thanks for your comments. The target trajectory is denoted by $T_{tar} =$ {$T_{tar}^1, ..., T_{tar}^K$}, $T_{tar}^i \in SE(3)$
> > , where $K$ is the length of the total sequence. We have revised the manuscript accordingly in Lines 246-247 marked with blue color.
> >
> > $P_{\text{obj}}$ denotes the point cloud of the target object. Specifically, $P_{\text{obj}} \in \mathbb{R}^{3 \times l}$ is a set of $l$ 3D coordinates $(x, y, z)$, where $l$ is the number of points in the point cloud. We have revised the manuscript accordingly in Lines 253-255 marked with blue color.
> >
> >
> >
> > > Q4:Is the target trajectory the 6D pose trajectory of an object or the 3D positions of a keypoint?
> >
> > A4: The target trajectory is 6D pose trajectory.

---

### Author Response · Authors · 2025-12-03
**General Response**

Dear PCs, SACs, ACs, and Reviewers,

We sincerely appreciate your efforts and contributions. Below, we provide a brief summary of the main points during the rebuttal period.


We are encouraged that **three reviewers express positive views of UniHM** (score 6, 6, 8). They find our paper:
- **Cross-embodiment dexterous manipulation is quite a contribution to the community** (Reviewer s9Mv, 7Tpz, ZSGv, uKvV)
- **Training purely from human data, providing a positive signal for addressing the issue of expensive data collection** (Reviewer s9Mv, uKvV)
- **The proposed approach is well-defined and has been validated to be viable.** (Reviewer 7Tpz, uKvV)
- **Powerful performance in evalution metrics cross multiple datasets** (Reviewer s9Mv, 7Tpz, ZSGv, uKvV)



---

During the rebuttal period, Reviewer uKvV (score 6) keep his/her positive score to support the accept of this paper, acknowledging that his/her concerns have been solved. The Reviewer s9Mv (score 6) and 7Tpz (score 8) also have positive scores. The only negative Reviewer ZSGv (score 2) did not have chance to respond to our rebuttal, but we believe our rebuttals have fully addressed all his/her concerns:

---

> Tokenizer evaluation limited to a single hand. Both the HOI and real-world experiments appear to use only one hand type.（Reviewer ZSGv)

We clarify that we actually used **5 types of hands in HOI experiments and 2 types of hands in real-world experiments**, and note the result in our appendix and supplementary material. Then, we added real-world experiments on a new high-DoF dexterous hand and provide new result video `cross_embodiment.mp4`.

---

> Concern about Evalution Metrics: i) Underspecified sequence-generation metrics. ii) Missing text–motion alignment evaluation. （Reviewer ZSGv)

For underspecified metrics, we note that these metrics are widely used and added the detailed explanation into our Appendix.
For missing metrics, we first noted that we report **FID as the quantitative evaluation** metric, then add a user study as qualitative analysis in our rebuttal.



> Further ablations and Comparative experiment （Reviewer s9Mv, uKvV)

We have added ablation studies for **different parts of the cost function**, an ablation on **MANO as a unified latent space**, additional comparisons with **Text2HOI** and **Grasp as You Say**, and the experiments on grasp **success rates** in simulation

---


> Time cost and Real-world set up of physical optimization（Reviewer s9Mv, uKvV)

We first note the **real-time** efficiency of our optimization method: it requires only **8.7 ms** to optimize per pose. We then discuss the robustness of the **robust kernel function** used in our optimization and added detailed explanation, a toy example and supplementary visualizations in the appendix.

---

> Complexity of the method, many models need to be finetuned.（Reviewer 7Tpz)

We note that our framework partially relies on the pre-trained and zero-shot capabilities of the VLM. However, this design enables our model to achieve powerful performance **without requiring expensive teleoperation data collection**; we only need to train on existing HOI datasets.

---

UniHM is the first dynamic and language-guided unified hand manipulation framework with a vision-language model. The positive reviewer feedback, rigorous methodology, and extensive experiments collectively show that UniHM offers a meaningful advancement in robot manipulation. We believe UniHM will help drive progress in fine-grained manipulation skills, vision–language action, and embodied interaction.

Sincerely,

The Authors

---

### Meta-Review · Area_Chair_3tfJ · 2026-01-07

**Summary:**

The paper got mixed ratings (6, 8, 2, 6).

Reviewers recognize the contribution to cross-embodiment dexterous manipulation and appreciate the superior performance demonstrated in the evaluations.

The reviewers’ concerns include unclear methodological details and evaluation metrics, as well as insufficient ablation studies and limited evaluation of certain aspects, such as text-motion alignment.

**Reviewer Concerns:**

The authors’ responses address most of the reviewers’ concerns.

The authors provide additional experiments, including expanded ablation studies, text–motion alignment evaluations, success rate metrics, and comparisons with other language-guided HOI models.

**Reviewer Scores:**

The AC expects that the positive ratings from reviewers remain unchanged and that the negative rating may increase.

The authors provide convincing evidence to support the strengths of their method and respond effectively to the concerns raised by the reviewers.

One reviewer with a positive rating (uKvV, rating 6) explicitly states that the concerns are resolved and continues to support the paper.

Importantly, the AC carefully checks the comments from the reviewer with the negative rating (ZSGv, rating 2) alongside the authors’ responses and finds that the responses are sufficiently convincing, with the expectation that the reviewer’s concerns can be addressed.

Based on these considerations, the AC recommends acceptance of this paper.

---

### Decision · Program_Chairs · 2026-01-26

Accept (Poster)